# Langevin Rollout Optimization for Modelic Reinforcement Learning

**Tianyi Zhang** [* 1]   **Likun Wang** [* 1]   **Guojian Zhan** [* 1]   **Feihong Zhang** [1]   **Yang Guan** [1]   **Yao Lyu** [1]
**Shengbo Eben Li** [1 2]

## Abstract

Planning-driven model-based (modelic) reinforcement learning has achieved impressive success in continuous control tasks but predominantly relies on zero-order optimizers like Model Predictive Path Integral (MPPI). While robust for global exploration, MPPI updates actions solely through sampling and neglects the reliable return gradients inherent in structured dynamics that guide fine-grained search. To complement MPPI's robustness with gradient-guided precision, we first propose **La**ngevin **R**ollout **O**ptimization (LaRO), which leverages return gradients to refine actions via Langevin dynamics, achieving reliable local convergence without sacrificing multimodal exploration. This is supported by a score-augmented world model that jointly learns dynamics and a score function within a unified latent space, facilitating efficient and accurate gradient estimation for real-time planning. Second, we combine MPPI and LaRO through a simple yet effective choice mechanism, termed **M**aximum **L**ook-**A**head **P**lanning (MLAP). Finally, we instantiate MLAP within the latest BOOM algorithm, replacing its MPPI-only planner and yielding BOOM-L. Empirical results on the DeepMind Control Suite and Humanoid Bench demonstrate that BOOM-L consistently outperforms strong baselines in both sample efficiency and final performance.

## 1. Introduction

Model-based reinforcement learning (MBRL) has achieved remarkable success across a diverse array of domains, ranging from strategic board games (Silver et al., 2016; Schrittwieser et al., 2020; Hafner et al., 2025) to high-dimensional control tasks in autonomous driving and embodied intelligence (Wu et al., 2022; Sikchi et al., 2022; Wu et al., 2023). Among various MBRL approaches, planning-driven methods are particularly prominent, leveraging learned dynamics for online planning to enable long-term reasoning and dynamic action refinement, which ultimately results in a superior policy (Schrittwieser et al., 2021; Kaiser et al., 2020). This characteristic empowers MBRL to surpass trial-and-error based model-free methods, particularly when tackling complex continuous control tasks (Curi et al., 2020).

To further scale the performance of planning-driven MBRL, recent research has focused on tighter integration between online planning and policy learning (Hamrick et al., 2020). A landmark work is the TD-MPC family (Hansen et al., 2022; 2024), which leverages high-quality trajectories generated by the planner to jointly optimize the dynamics, reward, and value functions via temporal difference learning, achieving strong performance across diverse challenging benchmarks. However, learning policy from planner-collected data inevitably induces a distribution shift issue known as actor divergence. Recent advancements such as LOOP (Sikchi et al., 2022), BMPC (Wang et al., 2025a), and BOOM (Zhan et al., 2026) mitigate this by constraining the planner-policy discrepancy through techniques like bootstrap alignment, further elevating performance.

Despite these successes, the online planner in these methods is predominantly implemented via a zero-order optimization technique, Model Predictive Path Integral (MPPI). Although MPPI supports broad exploration, it treats the differentiable world model as a black-box objective and refines actions solely through sampling, thereby discarding informative model gradients and often *leading to slow convergence and oscillatory updates on sharp landscapes* (Williams et al., 2017). This oversight is particularly critical in MBRL, since the learned dynamics and reward models are constrained by stationary physical laws and thus often admit smooth, reliable gradients, providing a strong signal to guide the search direction. Consequently, existing MBRL algorithms fail to fully harness the world model's optimization potential, fundamentally limiting their achievable performance.

In this paper, we propose **La**ngevin **R**ollout **O**ptimization (LaRO), which refines actions using gradient-guided

---

[*]Equal contribution [1]School of Vehicle and Mobility, Tsinghua University, Beijing, China [2]College of Artificial Intelligence, Tsinghua University, Beijing, China. Correspondence to: Shengbo Eben Li <lishbo@tsinghua.edu.cn>.

*Proceedings of the 43rd International Conference on Machine Learning*, Seoul, South Korea. PMLR 306, 2026. Copyright 2026 by the author(s).

Langevin dynamics, following high planning-return directions while still traversing energy barriers to explore multimodal solutions. We further learn an auxiliary score function that captures the intrinsic optimization field within a unified latent space shared with the world model, enabling accurate return gradient estimation while bypassing the computational burden of backpropagation through time (BPTT) during planning. To couple broad exploration with precise refinement, we introduce a hybrid planning framework, named **M**aximum **L**ook-**A**head **P**lanning (MLAP). This framework runs MPPI and LaRO branches in parallel, dynamically executing the candidate with the higher look-ahead return to synergize global coverage with local optimization.

Specifically, our contributions can be summarized as follows: **(1)** We propose *LaRO*, which is the first to introduce Langevin dynamics into online planning for MBRL. To support LaRO, we design a score-augmented world model to enable efficient look-ahead return gradient computation. **(2)** We unify LaRO and MPPI under the *MLAP* framework to synergize local precision with global coverage, and instantiate it within the latest planning-driven MBRL algorithm, BOOM (Zhan et al., 2026), yielding *BOOM-L*. **(3)** Extensive experiments on high-dimensional continuous control benchmarks, including the DeepMind Control Suite (Tassa et al., 2018) and Humanoid Bench (Sferrazza et al., 2024), demonstrate that BOOM-L achieves consistent improvements over BOOM and other baselines.

## 2. Related Works

### 2.1. Model-based Reinforcement Learning

MBRL leverages a learned world model, typically comprising a dynamics model and a reward model, to facilitate efficient policy optimization and decision-making (Li, 2023; Schrittwieser et al., 2020; Hafner et al., 2025; Hansen et al., 2022; Wang et al., 2025a). Existing MBRL methods are commonly grouped into *planning-driven* and *imagination-driven* paradigms according to the role the world model plays in training.

*Imagination-driven* methods use the world model to produce synthetic rollouts, which augment real experience and enable additional policy and value updates, thereby substantially improving sample efficiency (Sutton, 1991; Kaiser et al., 2020; Micheli et al., 2023). The Dreamer family (Hafner et al., 2020; 2021; 2025) stands as a prominent representative of modern imagination-driven approaches, which typically learn compact latent dynamics to perform actor-critic updates entirely within the latent imagination. In particular, DreamerV3 (Hafner et al., 2025) further scales world-model learning and achieves consistently strong performance across diverse domains. Nevertheless, these meth-

ods are limited by compounding model errors over long imagined horizons and often require substantial environment interaction to converge.

*Planning-driven* methods leverage the learned world model for explicit online planning to derive the better behavior policy for environment interaction, often achieving strong performance with fewer interaction steps (Curi et al., 2020; Janner et al., 2019; Lin et al., 2025). Contemporary methods typically incorporate a learned policy to warm-start the planner, accelerating the search process and guiding it towards promising high-value regions (Wang & Ba, 2020; Nguyen et al., 2021; Sikchi et al., 2022; Zhan et al., 2026). For instance, LOOP (Sikchi et al., 2022) integrates a finite-horizon planner into the SAC framework to enhance exploration, and the TD-MPC family (Hansen et al., 2022; 2024) utilizes a terminal value function to ground short-horizon planning within a learned latent space. These methods substantially outperform model-free baselines in challenging continuous control domains. Recent works, such as BMPC (Wang et al., 2025a) and BOOM (Zhan et al., 2026), have further improved final performance by addressing the actor divergence issue to ensure strict consistency between the planner and the policy. However, these methods predominantly rely on MPPI, which neglects valuable gradient information of world models and often struggles to converge on sharp landscapes. In contrast, our method introduces Langevin dynamics to actively leverage these gradients, ensuring fine-grained action refinement.

### 2.2. Online Planner

Online planning algorithms serve as the decision-making core in MBRL, enabling the agent to select optimal actions by simulating future trajectories within a world model. Unlike offline policy optimization, online planners perform Model Predictive Control (MPC) to solve a finite-horizon optimization problem at each time step.

The most prevalent class of online planners is sampling-based MPC, which transforms the trajectory optimization problem into a black-box optimization task. Random Shooting (RS) represents the simplest form, where multiple action sequences are sampled uniformly and the one with the highest predicted return is executed. However, RS often suffers from low sample efficiency in high-dimensional action spaces. To address this, the Cross-Entropy Method (CEM) is widely adopted in frameworks like PETS (Chua et al., 2018) and PlaNet (Hafner et al., 2019). CEM iteratively refines a sampling distribution (typically a diagonal Gaussian) by fitting it to the "elite" samples that yield the highest rewards. While CEM is effective, it can be sensitive to local optima and population size. To achieve smoother and more robust control, researchers have introduced Information-Theoretic MPC methods, most notably MPPI control (Williams et al.,

2017). MPPI utilizes a soft-max weighted average of all sampled trajectories based on their cumulative costs, which allows for better exploration of the state space and is highly parallelizable on modern hardware.

These traditional sampling-based planners are fundamentally zero-order optimizers. By treating the world model effectively as a black-box evaluator, they ignore the rich gradient information inherent in differentiable models. Conversely, recent approaches have integrated first-order derivatives to optimize actions via Gradient Descent (Henaff et al., 2017; Jyothir et al., 2023). However, pure gradient optimization is greedy and deterministic, making it prone to local optima and ill-suited for multimodal control. This motivates Langevin dynamics, which adds noise to gradient updates to retain gradient precision while enabling stochastic, multimodal exploration.

# 3. Preliminaries

## 3.1. Reinforcement Learning

Reinforcement learning (RL) formulates sequential decision-making as a Markov Decision Process (MDP) defined by the tuple $(\mathcal{S}, \mathcal{A}, P, r, \gamma)$, where $\mathcal{S}$ is the state space, $\mathcal{A}$ is the action space, $P(s'|s, a)$ denotes the transition dynamics, $r(s, a)$ is the reward function, and $\gamma \in [0, 1)$ is the discount factor (Li, 2023; Lyu et al., 2025). At each time step $t$, the agent observes the current state $s_t \in \mathcal{S}$, selects an action $a_t \in \mathcal{A}$, receives a reward $r_t = r(s_t, a_t)$, and transitions to the next state $s_{t+1} \sim P(\cdot|s_t, a_t)$. The agent's behavior is determined by a policy $\pi(a|s)$, which maps states to a distribution over actions.

The goal of RL is to find an optimal policy $\pi^*$ that maximizes the expected cumulative discounted return: $J(\pi) = \mathbb{E}_\pi \left[ \sum_{t=0}^\infty \gamma^t r(s_t, a_t) \right]$. A central concept in RL is the action-value function (Q-function) $Q^\pi(s, a)$, which estimates the expected return starting from state $s$, taking action $a$, and thereafter following policy $\pi$: $Q^\pi(s, a) = \mathbb{E}_\pi \left[ \sum_{k=0}^\infty \gamma^k r(s_{t+k}, a_{t+k}) \mid s_t = s, a_t = a \right]$.

## 3.2. Online Planning

Online planning optimizes a short-horizon action sequence at every environment step by rolling out predicted trajectories in a world model, then executing the first action of the best plan. By explicitly reasoning over future outcomes, it often produces higher-quality decisions than a purely reactive policy, and is therefore widely used in MBRL to collect stronger interaction data and improve policy performance. Formally, given a state $s_t$, a dynamics model $s_{t+1} = f(s_t, a_t)$, a reward model $r_t = R(s_t, a_t)$, a terminal value function $q_t = Q(s_t, a_t)$ and a planning horizon $H$, the planner searches for an action sequence $\mathbf{a}_t = a_{t:t+H-1}$

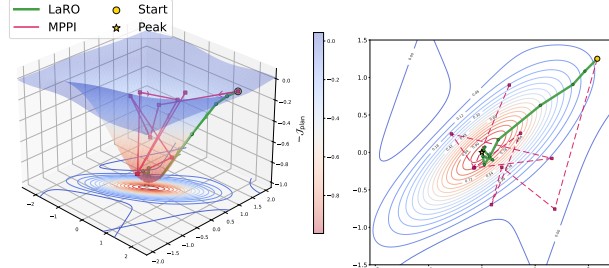

*Figure 1.* **Toy planning experiment.** LaRO (green) rapidly converges to the peak under gradient guidance, while MPPI (magenta) exhibits oscillatory updates; the left panel shows the 3D return landscape and the right panel shows the corresponding 2D contours and trajectories.

that maximizes the predicted discounted return:

$$\mathcal{J}_{\text{plan}}(s_t, \mathbf{a}_t) = \gamma^H q_{t+H} + \sum_{h=0}^{H-1} \gamma^h r_{t+h}. \quad (1)$$

Among many online planners, Model Predictive Path Integral (MPPI) is a particularly popular choice for continuous control due to its simplicity and efficiency (Williams et al., 2017; Hansen et al., 2022; 2024). At each step, MPPI samples $N$ candidate action sequences $\{\mathbf{a}_t^i\}_{i=1}^N$ from a Gaussian distribution $\mathcal{N}(\boldsymbol{\mu}_t, \boldsymbol{\sigma}_t^2)$, performs model rollouts to evaluate $\mathcal{J}_{\text{plan}}^i$ for each trajectory, and then updates the sampling mean via return-weighted averaging: $\boldsymbol{\mu}_t \leftarrow \sum_i w_t^i \mathbf{a}_t^i$, where the normalized weights are $w_t^i = \text{softmax}(\mathcal{J}_{\text{plan}} - \max_j \mathcal{J}_{\text{plan}}^j)_i$. Repeating this procedure for a few iterations yields an improved plan, after which the first action is applied in a receding-horizon manner.

## 3.3. Langevin Dynamics

Langevin dynamics (Welling & Teh, 2011) constitutes a powerful framework for drawing samples from a complex probability distribution $p(\cdot)$ by leveraging its score function $\nabla_x \log p(x)$. Starting from an initial state sampled from a prior distribution $x_0 \sim \rho(x)$, the method iteratively refines the samples following:

$$x_{t+1} = x_t + \eta \nabla_{x_t} \log p(x_t) + \sqrt{2\eta} \epsilon \quad (2)$$

where $\eta$ denotes the step size, and $\epsilon \sim \mathcal{N}(\mathbf{0}, \mathbf{I})$ represents standard Gaussian noise. Theoretically, as the step size $\eta \to 0$ and the number of iterations $t \to \infty$, the distribution of $x_t$ converges to $p(\cdot)$ (Song & Ermon, 2019).

# 4. Method

## 4.1. Enhanced Planning via Langevin Dynamics

Although the MPPI reviewed in Sec 3.2 has demonstrated strong empirical performance across a wide range of control

tasks, such as locomotion (Hansen et al., 2024) and navigation (Roth et al., 2025), it is fundamentally a zero-order optimizer. Specifically, MPPI refines candidate action sequences through sampling and return reweighting, without explicitly exploiting gradient information available from differentiable world models and value functions. As a consequence, when the action-return landscape is sharp, MPPI can be sensitive to the sampling budget and may exhibit slow convergence or unstable behavior driven by stochastic sampling without gradient guidance. As shown in Figure 1, in a toy experiment with an objective featuring a sharp peak and complex surrounding geometry (see Appendix B.1 for the details), MPPI iterates fluctuate across nearby ridges and valleys instead of consistently concentrating around the optimum, suggesting that sampling-and-reweighting alone may lead to oscillatory and unreliable search on such sharp and rugged landscapes.

Motivated by this observation, we aim to design an online planner that (i) is *gradient-informed* to enable accurate local refinement on sharp landscapes, while (ii) still *maintains stochasticity* to represent and explore multiple promising action modes. To this end, we propose **La**ngevin **R**ollout **O**ptimization (LaRO), which achieves both objectives by incorporating Langevin dynamics into the online planning process. Specifically, we view online planning as inference over a planning policy $\pi_{\text{plan}}(a|s)$ and introduce a prior policy $\rho(a|s)$ that provides a task-agnostic proposal and supports exploration. Inspired by the maximum-entropy RL principle (Haarnoja et al., 2018; Duan et al., 2025), at each state $s_t$, we define the planning objective as

$$\max_{\pi_{\text{plan}}} \mathbb{E}_{a_t \sim \pi_{\text{plan}}(\cdot|s_t)} \left[ \mathcal{J}_{\text{plan}}(s_t, a_t) - \alpha \log \pi_{\text{plan}}(a_t|s_t) \right], \quad (3)$$

where $\alpha > 0$ is the temperature parameter controlling the stochasticity of the planning policy, and $\mathcal{J}_{\text{plan}}(s_t, a_t)$ denotes the $H$-step look-ahead return in (1) induced by committing to the first action $a_t$ while subsequent actions are generated by rolling out the dynamics model with the prior policy as $a_{t+1} \sim \rho(\cdot|f(s_t, a_t))$; see Remark A.9 for the motivation. Under (3), the optimal planning policy takes a Boltzmann form (Psenka et al., 2024):

$$\pi_{\text{plan}}^*(a|s) = \frac{1}{Z(s)} \exp \left( \frac{\mathcal{J}_{\text{plan}}(s, a)}{\alpha} \right), \quad (4)$$

where $Z(s)$ is the normalizing constant. Importantly, this distribution can be multimodal when $\mathcal{J}_{\text{plan}}$ has multiple high-return regions, thereby naturally capturing diverse action hypotheses (Psenka et al., 2024; Wang et al., 2025b; Ishfaq et al., 2025). To draw samples from the optimal distribution, we apply Langevin dynamics illustrated in Sec. 3.3 in the action space. Noting that $\nabla_a \log \pi_{\text{plan}}^*(a|s) = \nabla_a \mathcal{J}_{\text{plan}}(s, a)/\alpha$, the planner action can be generated iteratively via

$$a_t^{(k+1)} = a_t^{(k)} + \frac{\eta}{\alpha} \nabla_a \mathcal{J}_{\text{plan}}(s_t, a_t^{(k)}) + \sqrt{2\eta}\epsilon. \quad (5)$$

In contrast to MPPI, the gradient term steers samples directly toward high-return regions, enabling sharp and directed improvement on peaked landscapes, while the injected Gaussian noise prevents premature collapse and allows the sampler to move across different basins, thereby preserving multimodality. Figure 1 shows that LaRO rapidly converges to the peak thanks to gradient guidance.

### 4.2. LaRO with Score-Augmented World Model

In RL tasks, the true environment dynamics and reward functions are opaque, preventing the direct computation of return gradients $\nabla_a \mathcal{J}_{\text{plan}}$ required for LaRO. Furthermore, even with learned dynamics and reward models, computing these gradients via Backpropagation Through Time (BPTT) during online planning is computationally expensive and hinders real-time performance. To address both challenges, we propose a unified world model that jointly learns to represent the environment dynamics and its associated optimization landscape.

Adopting the architectural paradigm of the TD-MPC2 (Hansen et al., 2024), we employ a latent space formulation. Uniquely, in addition to the standard components including latent state encoder $z = h(s)$, latent dynamics model $z' = f(z, a)$, reward predictor $R(z, a)$, and value function $Q(z, a)$, we introduce an amortized score network $S(z, a)$ as an intrinsic component of the world model. This network is designed to approximate the gradient of the planning objective, effectively encoding the optimization field directly within the latent space. All components are optimized jointly by minimizing a unified objective:

$$\mathcal{L}_{\text{total}} = \mathbb{E}_{(s,a,r,s')_{0:H} \sim \mathcal{B}} \left[ \mathcal{J}_{\text{score}} + \sum_{t=0}^{H} \mathcal{J}_{\text{model},t} \right],$$

$$\mathcal{J}_{\text{score}} = \|\alpha S(z_0, a_0) - \text{sg}(\nabla_a \mathcal{J}_{\text{plan}}(z_0, a_0))\|_2^2, \quad z = h(s),$$

$$\begin{aligned} \mathcal{J}_{\text{model},t} = {} & \lambda_1 \|f(z_t, a_t) - \text{sg}(z_{t+1})\|_2^2 \\ & + \lambda_2 \text{CE}(R(z_t, a_t), r_t) \\ & + \lambda_3 \text{CE}(Q(z_t, a_t), r_t + \gamma Q_-(z_{t+1}, \pi(z_{t+1}))) \end{aligned}$$

$$(6)$$

where $\mathcal{B}$ is the replay buffer, and $(s, a, r, s')_{0:H}$ represents a trajectory segment of horizon $H$ sampled from $\mathcal{B}$ with batch size $N$. Here, $\text{sg}(\cdot)$ denotes the stop-gradient operator, $\text{CE}(\cdot, \cdot)$ is the cross-entropy loss, $Q_-$ is the target value network and $\pi(\cdot)$ is the policy network. The term $\mathcal{J}_{\text{score}}$ represents an explicit score matching loss, where the score network distills the analytical return gradients $\nabla_a J_{\text{plan}}$ derived by unrolling the model components $f$, $R$ and $Q$. By integrating this objective, the latent representation is shaped not only to predict reward and future states but also to facilitate accurate gradient estimation, thereby enhancing the model's utility for planning. At inference time, $S(\cdot, \cdot)$ replaces the expensive BPTT unrolling, enabling rapid, low-

latency Langevin updates as

$$a_t^{(k+1)} = a_t^{(k)} + \eta S(h(s_t), a_t^{(k)}) + \sqrt{2\eta}\epsilon. \quad (7)$$

We theoretically prove that, given a bounded score approximation error, the action distribution generated by this update rule will converge to a neighborhood of the optimal planning distribution.

**Theorem 4.1** (Convergence with Score Function). *Let the potential function $-\log \pi_{plan}^*$ be $\lambda$-strongly convex (see Remark A.10 for a discussion on this assumption in multimodal landscapes). If the score estimation error is bounded by $\mathbb{E}[\|S - \nabla \log \pi_{plan}^*\|^2] \leq \delta^2$, then the Wasserstein distance between the target $\pi_{plan}^*$ and the distribution $q_k$ induced by $a_t^{(k)}$ satisfies $\limsup_{k \to \infty} W_2(q_k, \pi_{plan}^*) \leq \delta/\lambda$.*

*Proof.* See Appendix A.2. □

Finally, we offer another critical insight into our method from the perspective of model-based versus model-free Langevin dynamics. Prior works applying Langevin dynamics to RL are predominantly model-free (Psenka et al., 2024; Ishfaq et al., 2025), updating actions via $\nabla_a Q$. Consequently, the quality of these updates is dominated by value function approximation errors, which are frequently amplified by bootstrapping and training non-stationarity. In contrast, our model-based Langevin planner computes $\nabla_a \mathcal{J}_{plan}$ by unrolling an $H$-step trajectory through a learned world model. The dominant gradient contributions stem from the supervised components, the dynamics $f$ and reward $R$, which are governed by stationary physical laws and typically converge to a low-error regime. Meanwhile, the value term $Q$ appears only at the end of the rollout, and its gradient contribution is down-weighted by the discount factor $\gamma^H$. As a result, the model-based return gradient can be more accurate and stable than directly using the value gradient, leading to more reliable Langevin updates. We theoretically prove that, provided model errors are smaller than value errors, the model-based return gradient achieves a lower worst-case error bound than its model-free counterpart.

**Theorem 4.2** (Model-based return gradients are more accurate than value gradients). *Let $\mathcal{J}(z_t, a_t)$ be the true $H$-step look-ahead return and $\hat{\mathcal{J}}(z_t, a_t)$ be its model-based counterpart induced by $\hat{f}, \hat{R}, \hat{Q}$. Under Assumptions A.5–A.8, the return-gradient error satisfies $\sup\|\nabla_a \hat{\mathcal{J}} - \nabla_a \mathcal{J}\| < \sup\|\nabla_a \hat{Q} - \nabla_a Q\|$.*

*Proof.* See Appendix A.3. □

### 4.3. Reinforcement Learning with Maximum Look-Ahead Planning

In this section, we further integrate the proposed LaRO into an off-policy RL framework to enhance policy performance.

---

**Algorithm 1 BOOM-L**

**Input:** Encoder $h_\epsilon$, dynamics $f_\xi$, reward $R_\omega$, value $Q_\phi$, score $S_\psi$, policy $\pi_\theta$, replay buffer $\mathcal{B}$,
**Initialize:** $h_\epsilon, f_\xi, R_\omega, Q_\phi, S_\psi, \pi_\theta, \mathcal{B}$

1: *// 1. Warmup world model with random actions*
2: $(r, s', \text{done}) \leftarrow \text{env.step}(a_{\text{random}})$
3: $\mathcal{B} \leftarrow \mathcal{B} \cup \{(s, a_{\text{random}}, r, s')\}$
4: Minimize $\mathcal{L}_{\text{total}}$ in (6) to update $h_\epsilon, f_\xi, R_\omega, Q_\phi, S_\psi$

5: **while** not converged **do**
6:   *// 2. Experience collection via MLAP*
7:   Encode current observation: $z = h_\epsilon(s)$
8:   $a_{\text{mppi}} \leftarrow \text{MPPI}(z_t, f_\xi, R_\omega, Q_\phi, \pi_\theta)$
9:   $a_{\text{lang}} \leftarrow \pi_\theta(\cdot|z_t)$
10:   **for** each LaRO iteration **do**
11:     Update $a_{\text{lang}}$ via (7)
12:   **end for**
13:   $a_{\text{plan}} \leftarrow \arg\max_a(\mathcal{J}_{\text{plan}}(z_t, a_{\text{mppi}}), \mathcal{J}_{\text{plan}}(z_t, a_{\text{lang}}))$
14:   $(r, s', \text{done}) \leftarrow \text{env.step}(a_{\text{plan}})$
15:   $\mathcal{B} \leftarrow \mathcal{B} \cup \{(s, a_{\text{plan}}, r, s')\}$

16:   *// 3. Update networks*
17:   **for** num updates per episode **do**
18:     Sample rollout batch $(s, a_{\text{plan}}, r, s')_{0:H} \sim \mathcal{B}$
19:     Minimize $\mathcal{L}_{\text{total}}$ in (6) to update $h_\epsilon, f_\xi, R_\omega, Q_\phi, S_\psi$
20:     Minimize $\mathcal{L}_{\text{policy}}$ in (8) to update $\pi_\theta$
21:   **end for**
22: **end while**

---

At each environment step, the RL network policy $\pi$ serves as the prior policy $\rho$ defined in Sec. 4.1 to initialize the planning process. Subsequently, the planner refines this proposal to produce higher-quality actions for interaction. This results in a replay buffer enriched with superior trajectories, which in turn accelerates policy learning and improves sample efficiency.

For policy optimization, we adopt a bootstrapped objective that balances maximization of the action value with alignment towards the high-quality actions generated by the planner. Specifically, we employ the training objective from BOOM (Zhan et al., 2026):

$$\mathcal{L}_{\text{policy}} = -\mathbb{E}_{s_{0:H} \sim \mathcal{B}}\left[\sum_{t=0}^{H-1} Q(z_t, \pi(z_t)) - \alpha \log \pi(z_t)\right]$$

$$- \lambda \mathbb{E}_{(s,a)_{0:H} \sim \mathcal{B}}\left[\sum_{i=0}^{N} w_i \log \pi(a_t^i | z_t^i)\right], \quad z = h(s)$$

$$(8)$$

where $w_i = \exp(Q_i/\tau)/\sum_{i=0}^{N}\exp(Q_i/\tau)$ is the weighting term over a batch of size $N$ sampled from the replay buffer. The first term represents the standard maximum entropy RL objective, encouraging the policy to maximize expected

returns while maintaining entropy for exploration. The second term functions as a likelihood-free alignment loss, which essentially minimizes the KL divergence between the planner policy and network policy to mitigate the actor divergence issue common in planning-driven MBRL.

For practical online planner implementation, we employ a competitive hybrid planning strategy, termed **M**aximum **L**ook-**A**head **P**lanning (MLAP), to ensure robust performance across diverse landscapes. Although our LaRO demonstrates superior efficiency and precision compared to MPPI, purely gradient-based refinement can theoretically be susceptible to local optima in highly non-convex landscapes. To address this, we utilize a dual-track approach where MPPI and LaRO operate in parallel. MPPI serves as a robust global search operator to maintain broad coverage of the action space, while LaRO utilizes the score function to precisely ascend local gradient peaks. A simple return-based selection mechanism then dynamically integrates these outputs by evaluating the look-ahead returns of candidates from both branches and executing the optimal one. This mechanism acts as a robust gatekeeper, combining the global exploration capability of zero-order sampling with the convergence precision of first-order optimization. Since our final MBRL algorithm retains BOOM's core design, namely the bootstrapped objective in (8), we refer to the resulting algorithm as **BOOM-L**. The pseudo-code of our BOOM-L is summarized in Algorithm 1.

# 5. Experiments

## 5.1. Experimental Setup

**Baselines.** We benchmark **BOOM-L** against five strong online RL baselines: (1) **SAC** (Haarnoja et al., 2018): the state-of-the-art (SOTA) off-policy, model-free maximum-entropy RL algorithm; (2) **DreamerV3** (Hafner et al., 2025): the SOTA imagination-driven MBRL algorithm; (3) **TD-MPC2** (Hansen et al., 2024): a representative planning-driven MBRL algorithm that performs online planning via MPPI; (4) **BMPC** (Wang et al., 2025a): an improved variant of TD-MPC2 that enhances policy learning through planner-guided imitation; (5) **BOOM** (Zhan et al., 2026), the latest planning-driven MBRL algorithm upon which our work is built, utilizing MPPI exclusively for planning. To ensure a fair comparison, we train all algorithms for 1M environment steps. Additionally, we evaluate DreamerV3 with an extended budget of 10M steps to fully demonstrate its asymptotic capabilities.

**Benchmarks.** We evaluate across 14 challenging, high-dimensional locomotion tasks from two benchmarks: the DeepMind Control Suite (DMC) (Tassa et al., 2018) and Humanoid Bench (H-Bench) (Sferrazza et al., 2024). The 7 DMC tasks involve the two most demanding agents in

the suite, humanoid (67/24 state/action dimensions) and dog (223/38), i.e., DMC-Hard, which require precise balance control and coordinated whole-body motion. The remaining 7 H-Bench tasks further increase difficulty with long-horizon, goal-conditioned locomotion on the Unitree H1hand robot (151/61), including scenarios such as traversing slides, moving through a pole forest without collisions, and repeatedly clearing hurdles. A full list of task specifications is provided in Appendix B.2. To further validate the generality of BOOM-L, we additionally evaluate on 21 tasks spanning three supplementary suites: DMC-Easy (7 tasks), Extra H-Bench with Hand (7 tasks), and H-Bench without Hand (7 tasks); results and analysis are provided in Appendix C.2.

**Implementation details.** The detailed hyperparameters and reproducibility statement of other baselines are documented in Appendix B.3.

## 5.2. Experimental Results

The training curves and detailed numerical comparisons are presented in Figure 2 and Table 1, respectively. Our BOOM-L consistently achieves the highest Total Average Return (TAR) across the full suite of 14 high-dimensional locomotion tasks.

**Results on the DMC Suite.** Our BOOM-L achieves an average TAR of **847.2**, surpassing the prior SOTA BOOM (802.7) by **+5.5%**, and significantly outperforming BMPC (704.4) and TD-MPC2 (504.0) by margins of **+20.3%** and **+68.1%**, respectively. The benefits of gradient-guided refinement are particularly evident in complex dynamic locomotion tasks, where zero-order sampling tends to stagnate on sharp return landscapes due to the lack of explicit directional guidance. For instance, in the most arduous *Humanoid-run* task, BOOM-L achieves a score of **510.3**, improving upon the second-best BOOM (446.2) by **+14.4%** and establishing a considerable margin.

**Results on the Humanoid Bench.** BOOM-L demonstrates even stronger dominance on this more challenging benchmark, achieving an average TAR of **773.3**, which exceeds BOOM (634.2) by a substantial **+21.9%** and outperforms the strong baseline **DreamerV3** (trained on 10M steps) by **+37.0%**. Most notably, in the notoriously hard *H1hand-run* task, the gradient-free MPPI in BOOM often gets trapped in poor local optima, achieving only 329.3, whereas BOOM-L leverages gradient guidance to escape these basins and reaches **823.0**. This yields a remarkable **+149.9%** improvement, underscoring MLAP's ability to uncover high-quality solutions in rugged landscapes where pure sampling breaks down. Additionally, in precision-demanding tasks such as *H1hand-slide* and *H1hand-hurdle*, BOOM-L further extends its lead over BOOM by **+25.4%**

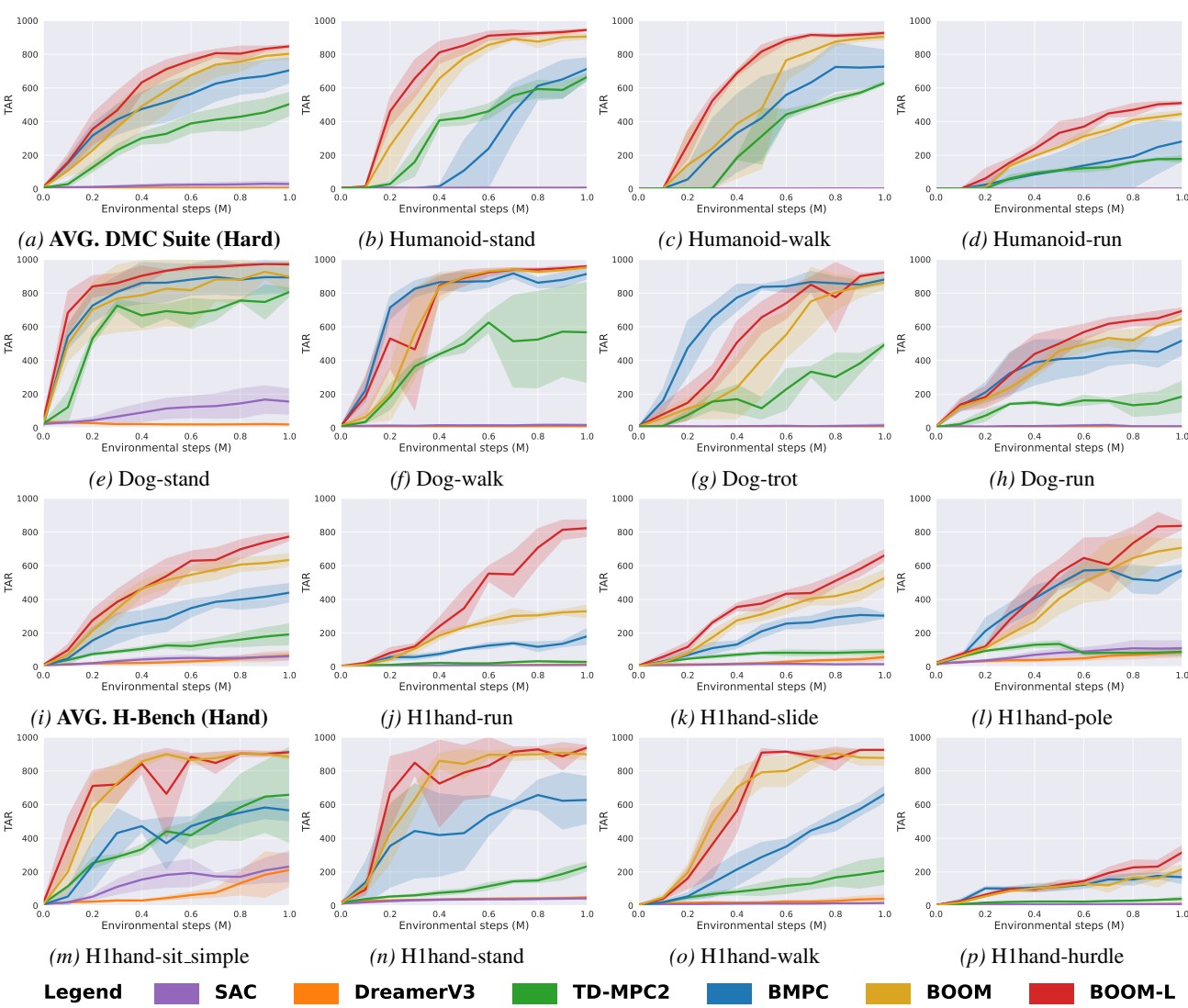

*Figure 2.* **Training curves on benchmarks.** The solid lines represent the mean, while the shaded regions indicate the confidence interval over five runs. The average performance curves for the two benchmarks appear at the left corner of the 1st and 3rd rows, respectively, highlighted in **bold**.

and **+45.8%**, respectively.

**Summary.** Model-free SAC and imagination-driven DreamerV3 struggle to reach strong performance within 1M steps, reflecting limited sample efficiency on these high-dimensional control tasks. In contrast, planning-driven baselines (TD-MPC2, BMPC, and BOOM) converge more reliably but often plateau, as they rely on zero-order sampling and underutilize the rich, accurate gradients available from the learned world model. By leveraging these gradients through the proposed MLAP, **BOOM-L** breaks this performance ceiling and achieves consistently higher final TAR across benchmarks.

### 5.3. Ablation Study

We conduct four ablation studies on the challenging *H1hand-run* task from H-Bench to quantify the contribution of each key component in our algorithm, as its high dimensionality and rugged reward landscape make it a stringent testbed.

**Score function.** To validate the fidelity of our score-augmented world model, we compare the proposed LaRO (using the learned score) against an oracle baseline that computes exact gradients via BPTT. As shown in Figure 3a, our method tracks the oracle closely with only a negligible gap in TAR. This confirms that the learned score accurately approximates the underlying look-ahead return gradients, effectively supporting gradient-based refinement.

*Table 1.* Total Average Return (TAR) on 7 DMC Suite tasks and 7 H-Bench tasks. Mean ± Std over 5 seeds. **Bold** = best, underlined = second-best. Higher is better.

| Task | SAC | DreamerV3 ( 1M & 10M steps) | | TD-MPC2 | BMPC | BOOM | BOOM-L |
|---|---|---|---|---|---|---|---|
| Humanoid-stand | $7.2 \pm 0.9$ | $4.4 \pm 0.5$ | $708.0 \pm 24.1$ | $665.5 \pm 23.2$ | $713.8 \pm 64.8$ | $905.3 \pm 16.3$ | **$944.6 \pm 6.6$** |
| Humanoid-walk | $1.9 \pm 0.4$ | $1.5 \pm 0.2$ | $745.8 \pm 27.6$ | $629.2 \pm 11.6$ | $726.4 \pm 98.9$ | $904.4 \pm 18.0$ | **$926.8 \pm 10.2$** |
| Humanoid-run | $1.2 \pm 0.1$ | $0.8 \pm 0.2$ | $343.1 \pm 34.0$ | $177.8 \pm 20.5$ | $282.5 \pm 116.9$ | $446.2 \pm 16.7$ | **$510.3 \pm 10.8$** |
| Dog-stand | $155.6 \pm 75.0$ | $19.8 \pm 2.1$ | $31.0 \pm 12.1$ | $806.9 \pm 24.5$ | $893.7 \pm 91.7$ | $897.9 \pm 94.6$ | **$971.0 \pm 5.7$** |
| Dog-walk | $16.6 \pm 2.8$ | $6.2 \pm 1.1$ | $7.9 \pm 2.1$ | $567.2 \pm 298.0$ | $914.3 \pm 30.4$ | $951.8 \pm 3.9$ | **$960.0 \pm 4.3$** |
| Dog-trot | $13.9 \pm 10.8$ | $6.2 \pm 1.3$ | $8.5 \pm 0.8$ | $495.3 \pm 14.3$ | $881.3 \pm 29.0$ | $865.0 \pm 40.5$ | **$923.1 \pm 5.8$** |
| Dog-run | $9.2 \pm 4.8$ | $3.4 \pm 2.0$ | $4.4 \pm 2.8$ | $186.1 \pm 91.9$ | $518.8 \pm 84.2$ | $648.2 \pm 32.8$ | **$694.7 \pm 17.5$** |
| **AVG. DMC Suite (Hard).** | $29.4 \pm 13.6$ | $6.0 \pm 1.0$ | $264.1 \pm 14.8$ | $504.0 \pm 69.1$ | $704.4 \pm 73.7$ | $802.7 \pm 31.8$ | **$847.2 \pm 8.7$** |
| H1hand-run | $9.1 \pm 1.4$ | $12.8 \pm 3.3$ | $649.1 \pm 74.0$ | $28.4 \pm 2.9$ | $180.3 \pm 47.7$ | $329.3 \pm 38.0$ | **$823.0 \pm 48.8$** |
| H1hand-slide | $14.8 \pm 2.4$ | $57.2 \pm 19.7$ | $356.3 \pm 32.3$ | $88.8 \pm 14.6$ | $303.0 \pm 15.0$ | $527.8 \pm 48.3$ | **$662.1 \pm 35.1$** |
| H1hand-pole | $109.1 \pm 45.1$ | $86.8 \pm 30.8$ | $562.4 \pm 59.9$ | $88.5 \pm 16.4$ | $571.7 \pm 33.8$ | $706.9 \pm 51.9$ | **$836.7 \pm 21.6$** |
| H1hand-stand | $42.7 \pm 6.1$ | $48.6 \pm 6.5$ | $844.5 \pm 23.7$ | $233.0 \pm 27.2$ | $627.4 \pm 138.5$ | $898.6 \pm 34.0$ | **$938.3 \pm 12.1$** |
| H1hand-walk | $15.1 \pm 2.5$ | $40.3 \pm 23.8$ | $741.4 \pm 25.3$ | $206.1 \pm 80.3$ | $662.4 \pm 48.5$ | $877.1 \pm 44.0$ | **$925.0 \pm 1.8$** |
| H1hand-sit_simple | $231.7 \pm 90.7$ | $211.5 \pm 96.2$ | $661.3 \pm 166.8$ | $658.2 \pm 283.6$ | $565.4 \pm 62.7$ | $882.9 \pm 14.7$ | **$911.5 \pm 6.7$** |
| H1hand-hurdle | $8.7 \pm 2.5$ | $10.0 \pm 2.8$ | $135.1 \pm 5.4$ | $40.2 \pm 12.6$ | $168.2 \pm 38.7$ | $216.9 \pm 25.7$ | **$316.3 \pm 40.6$** |
| **AVG. H-Bench (Hand).** | $61.6 \pm 21.5$ | $66.7 \pm 26.2$ | $564.3 \pm 55.3$ | $191.9 \pm 62.5$ | $439.8 \pm 55.0$ | $634.2 \pm 36.6$ | **$773.3 \pm 23.8$** |

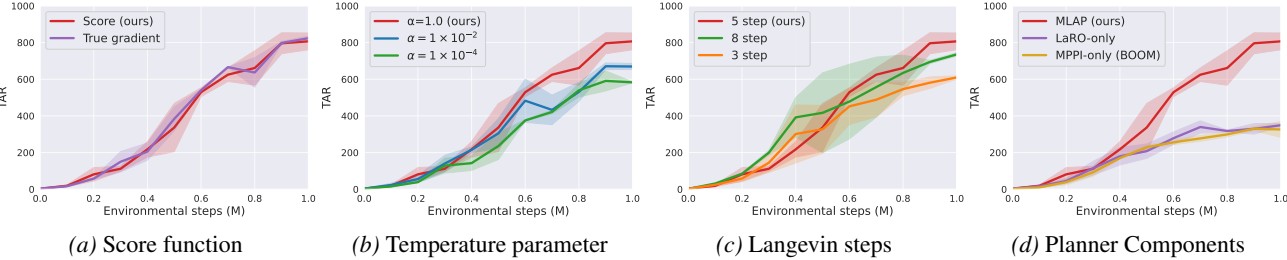

*(a)* Score function     *(b)* Temperature parameter     *(c)* Langevin steps     *(d)* Planner Components

*Figure 3.* **Ablation study curves.** We select the *H1hand-run* task (151/61) in H-Bench to perform all ablation experiments.

**Temperature parameter.** We analyze the impact of the temperature $\alpha$ in Figure 3b, which shows that performance drops as $\alpha$ decreases. Lower temperatures reduce the exploration noise, effectively degenerating LaRO into *deterministic gradient ascent*. This lack of stochasticity prevents the planner from escaping local optima in sharp, multimodal landscapes, resulting in inferior convergence.

**Langevin steps.** We analyze the impact of the number of Langevin steps in Figure 3c. We observe that fewer steps (3) result in under-performance due to insufficient convergence to the optima. Conversely, excessive steps (8) lead to slight degradation, likely caused by accumulated noise deviating actions from the optimal solution. A moderate setting (5) strikes the best balance between precision and stability.

**Planner Components.** Figure 3d compares the performance of individual planner components. We observe that LaRO-only performs similarly to the MPPI-only baseline, indicating that gradient refinement alone cannot compensate for the lack of global exploration. In contrast, MLAP effectively integrates both strengths, achieving a distinct "$1 + 1 > 2$" synergy that significantly outperforms either

component in isolation.

### 5.4. Efficiency Comparison

We conducted a comprehensive runtime efficiency analysis to evaluate the computational trade-offs of our proposed method. Table 2 provides a detailed breakdown of the inference latency for each module within our MLAP framework on the *H1hand-run* task. The results reveal that the primary computational bottlenecks during inference are the iterative trajectory sampling in MPPI and the look-ahead rollouts required for final action selection (Eval.& Argmax). Notably, with the introduction of the amortized score function, LaRO incurs a negligible latency of only 1.79 ms, adding minimal overhead to the planning loop. This represents a substantial speedup of over $7\times$ compared to the BPTT-based refinement (LaRO w/o Score), validating the efficiency of our score-based gradient estimation.

Table 3 compares the average 1M-step wall-clock training time and inference latency evaluated across four tasks (*Dog-run*, *Humanoid-run*, *H1hand-slide*, *H1hand-run*). This highlights two key cost trade-offs: (i) BOOM-L vs. BOOM-L w/o Score: BOOM-L is significantly more efficient. The

BPTT-based LaRO requires $K$ BPTT passes per environment step ($K = 5$: Langevin iterations) during online planning. Our score-based LaRO achieves zero online BPTT by elegantly shifting these expensive operations entirely to the network update phase, where they are computed only once per batch. (ii) BOOM-L vs. BOOM: BOOM-L incurs slightly higher computational costs than the purely sampling-based BOOM. The training overhead stems from updating the score network, while the slight increase in inference latency ($\sim$6 ms) is caused by MLAP's look-ahead action selection.

*Table 2.* **Component-wise inference latency on *H1hand-run* task.** The breakdown shows that the score-based LaRO is significantly faster than the BPTT baseline.

| Component | Latency [ms] |
|---|---|
| LaRO (Ours, w/ Score) | 1.79 |
| LaRO (w/o Score) | 12.72 |
| MPPI | 28.87 |
| Selection (Eval. & Argmax) | 4.27 |

## 6. Conclusion

In this paper, we present BOOM-L, a MBRL algorithm that bridges the gap between zero-order sampling and first-order optimization. Prior planning-driven methods treat the world model as a black box, neglect the reliable gradients inherent in physics-governed dynamics, and often stagnate on sharp landscapes. To overcome this, we proposed LaRO, which leverages these gradients to achieve precise action refinement. LaRO is supported by our score-augmented world model, which captures the intrinsic optimization landscape within a unified latent space, enabling efficient, BPTT-free gradient estimation. We further integrated LaRO with MPPI under the MLAP framework, synergizing the robustness of global sampling with the precision of gradient-based search. Empirical results on the DeepMind Control Suite and Humanoid-Bench demonstrate that BOOM-L consistently outperforms strong baselines in both sample efficiency and asymptotic performance. Notably, our approach exhibits exceptional robustness in complex tasks where pure sampling methods struggle, validating that harnessing the full differentiability of the world model is critical for scaling MBRL to high-dimensional continuous control.

**Limitations and Future Work.** Our framework has two primary limitations worth discussing. First, gradient-guided refinement relies on a sufficiently accurate world model: in regions where the learned dynamics or reward contains localized inaccuracies, particularly early in training or in sparsely visited parts of the state-action space, the score estimate may point toward spurious extrema, and pure LaRO can greedily exploit such errors and stagnate in local optima. The MLAP

*Table 3.* **Computational efficiency analysis.** Comparison of total wall-clock training time (1M steps) and inference latency, averaged over four tasks (*Dog-run*, *Humanoid-run*, *H1hand-slide*, *H1hand-run*).

| Method | Training (1M) [h] | Inference [ms] |
|---|---|---|
| BOOM | 20.08 | 28.95 |
| BOOM-L (Ours) | 25.51 | 34.76 |
| BOOM-L w/o Score | 42.09 | 45.97 |

framework mitigates this by retaining MPPI's broad stochastic sampling as a parallel safeguard, but does not eliminate the risk entirely. Second, while MLAP achieves substantial performance gains on tasks with sharp, rugged return landscapes, it incurs a moderate computational overhead relative to purely zero-order planners. On simpler, smooth-landscape tasks where MPPI is already near the performance ceiling (e.g., several DMC-Easy tasks in Appendix C.2), the benefit of LaRO is naturally modest, making this overhead a less favorable trade-off. Future work will explore integrating epistemic uncertainty into the score network to penalize unreliable gradients explicitly, and developing adaptive triggering mechanisms such as the AHP variant in Appendix C.1 to invoke LaRO only when needed, dynamically balancing efficiency and precision.

## Acknowledgements

This work is supported by SunRisingAI Lab and Tsinghua-Efort Joint Research Center for EAI Computation and Perception.

## Impact Statement

This paper presents BOOM-L, which aims to advance the field of model-based reinforcement learning for high-dimensional control. By significantly improving sample efficiency and final performance through gradient-guided planning, our work has the potential to accelerate the development and deployment of embodied AI agents, such as humanoid robots, in complex industrial and service environments.

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

# A. Theoretical Analysis

## A.1. Useful Lemmas

**Lemma A.1** (Triangle Inequality). *For any vectors $x, y \in \mathbb{R}^n$, the triangle inequality states that the norm of the sum of two vectors is less than or equal to the sum of their norms:*

$$\|x + y\| \leq \|x\| + \|y\|,$$

*where $\|\cdot\|$ denotes the standard Euclidean norm.*

**Lemma A.2** (Operator Norm Bound). *For any matrix $M \in \mathbb{R}^{m \times n}$ and any vector $v \in \mathbb{R}^m$, the induced (operator) $\ell_2$-norm satisfies*

$$\|M^\top v\|_2 \leq \|M\|_2 \|v\|_2,$$

*where $\|M\|_2 \triangleq \sup_{\|x\|_2 = 1} \|Mx\|_2$ denotes the spectral (operator) norm. Equivalently,*

$$\|Mv\|_2 \leq \|M\|_2 \|v\|_2.$$

**Lemma A.3** (Wasserstein-2 Distance). *Let $\mathcal{P}_2(\mathbb{R}^d)$ be the space of probability measures with finite second moments. For any two distributions $\mu, \nu \in \mathcal{P}_2(\mathbb{R}^d)$, the Wasserstein-2 distance is defined as:*

$$W_2(\mu, \nu) \triangleq \left( \inf_{\gamma \in \Gamma(\mu, \nu)} \mathbb{E}_{(x,y) \sim \gamma} \left[ \|x - y\|_2^2 \right] \right)^{1/2},$$

*where $\Gamma(\mu, \nu)$ denotes the set of all couplings with marginals $\mu$ and $\nu$. Being a metric, it satisfies the triangle inequality for any third distribution $\rho$:*

$$W_2(\mu, \nu) \leq W_2(\mu, \rho) + W_2(\rho, \nu).$$

*Furthermore, for any specific pair of random variables $(X, Y)$ with marginals $X \sim \mu$ and $Y \sim \nu$ (i.e., a specific coupling), the distance is upper bounded by the root mean square error:*

$$W_2(\mu, \nu) \leq \sqrt{\mathbb{E}\left[ \|X - Y\|_2^2 \right]}.$$

**Lemma A.4** (Contraction of Exact Langevin Step). *Assume the potential energy function $\mathcal{E}(x) = -\log p^*(x)$ is $\lambda$-strongly convex. Let $q_k$ be the current distribution and $q_{k+1}$ be the distribution resulting from one step of exact Langevin dynamics with step size $\eta$. Under synchronous coupling, the Wasserstein-2 distance contracts as:*

$$W_2(q_{k+1}, p^*) \leq (1 - \lambda\eta)W_2(q_k, p^*) + O(\eta^{1.5}).$$

*For sufficiently small $\eta$, the process exhibits linear convergence determined by the geometry of the target distribution.*

*Proof.* To analyze the contraction property, we employ the technique of *synchronous coupling*. Let $x_k \sim q_k$ be a sample from the current distribution and $x_k^* \sim p^*$ be a sample from the target distribution. We couple the evolution of both particles by sharing the same Gaussian noise $\epsilon \sim \mathcal{N}(0, I)$.

The update rule for the variable starting from $q_k$ is:

$$x_{k+1} = x_k - \eta\nabla\mathcal{E}(x_k) + \sqrt{2\eta}\epsilon.$$

Since $p^*$ is the stationary distribution, a particle starting from stationarity remains approximately stationary. Thus, we write the coupled update for $x_k^*$:

$$x_{k+1}^* = x_k^* - \eta\nabla\mathcal{E}(x_k^*) + \sqrt{2\eta}\epsilon.$$

We consider the squared Euclidean distance between the coupled particles:

$$\begin{aligned}
\|x_{k+1} - x_{k+1}^*\|_2^2 &= \|(x_k - x_k^*) - \eta(\nabla\mathcal{E}(x_k) - \nabla\mathcal{E}(x_k^*))\|_2^2 \\
&= \|x_k - x_k^*\|_2^2 + \eta^2\|\nabla\mathcal{E}(x_k) - \nabla\mathcal{E}(x_k^*)\|_2^2 \\
&\quad - 2\eta\langle x_k - x_k^*, \nabla\mathcal{E}(x_k) - \nabla\mathcal{E}(x_k^*)\rangle.
\end{aligned}$$

Note that the noise terms $\sqrt{2\eta}\epsilon$ cancel out perfectly due to synchronous coupling.

By the assumption that $\mathcal{E}(x)$ is $\lambda$-strongly convex, we have the inequality:

$$\langle \nabla\mathcal{E}(u) - \nabla\mathcal{E}(v), u - v\rangle \geq \lambda\|u - v\|_2^2.$$

Applying this to the cross-term:

$$-2\eta\langle x_k - x_k^*, \nabla\mathcal{E}(x_k) - \nabla\mathcal{E}(x_k^*)\rangle \leq -2\eta\lambda\|x_k - x_k^*\|_2^2.$$

Substituting this back into the distance expansion and grouping terms:

$$\|x_{k+1} - x_{k+1}^*\|_2^2 \leq (1 - 2\lambda\eta)\|x_k - x_k^*\|_2^2 + \eta^2\|\nabla\mathcal{E}(x_k) - \nabla\mathcal{E}(x_k^*)\|_2^2$$
$$= (1 - 2\lambda\eta)\|x_k - x_k^*\|_2^2 + O(\eta^2).$$

Taking expectations on both sides and utilizing Lemma A.3 yields

$$W_2^2(q_{k+1}, p^*) \leq \mathbb{E}\left[\|x_{k+1} - x_{k+1}^*\|_2^2\right] \leq (1 - 2\lambda\eta)\mathbb{E}\left[\|x_k - x_k^*\|_2^2\right] + O(\eta^2).$$

Assuming the coupling between $x_k$ and $x_k^*$ is optimal for the previous step, i.e., $\mathbb{E}[\|x_k - x_k^*\|_2^2] = W_2^2(q_k, p^*)$, and taking the square root:

$$W_2(q_{k+1}, p^*) \leq \sqrt{1 - 2\lambda\eta}\,W_2(q_k, p^*) + O(\eta^{1.5}).$$

Using the first-order Taylor approximation $\sqrt{1 - z} \approx 1 - z/2$ for small $\eta$, we obtain the contraction bound:

$$W_2(q_{k+1}, p^*) \leq (1 - \lambda\eta)W_2(q_k, p^*) + O(\eta^{1.5}).$$

This completes the proof. $\qquad\square$

## A.2. Proof of Theorem 4.1

For notation simplicity, we denote the optimization action $a_t^{(k)}$ at step $k$ as $a_k$, the learned score function as $S(a)$, the planner policy $\pi_{\text{plan}}(a|z)$ as $\pi(a)$, and the planning objective $\mathcal{J}_{\text{plan}}(z, a)$ as $\mathcal{J}(a)$. Consequently, the target distribution is given by $\pi^*(a) \propto \exp(\mathcal{J}(a)/\alpha)$. Let $\nabla\log\pi^*(a)$ denote the true score function.

We analyze the convergence of the generated distribution $q_k$, the distribution induced by $a_k$, to the target $\pi^*$ under the assumption that the learned score has a bounded error:

$$\mathbb{E}_{a\sim q_k}[\|S(a) - \nabla\log\pi^*(a)\|_2^2] \leq \delta^2.$$

*Proof.* Let $a_k \sim q_k$ be the sample at step $k$. To bound $W_2(q_{k+1}, \pi^*)$, we introduce an auxiliary variable $a_{k+1}'$ representing one step of *exact* Langevin dynamics starting from the *same* $a_k$, coupled with the same noise $\epsilon_k \sim \mathcal{N}(0, I)$ used in the algorithm:

$$\text{LaRO Step:} \quad a_{k+1} = a_k + \eta S(a_k) + \sqrt{2\eta}\epsilon_k \tag{9}$$

$$\text{Ideal Step:} \quad a_{k+1}' = a_k + \eta\nabla\log\pi^*(a_k) + \sqrt{2\eta}\epsilon_k \tag{10}$$

Let $q_{k+1}'$ denote the distribution of $a_{k+1}'$. By the triangle inequality in Lemma A.3, we have

$$W_2(q_{k+1}, \pi^*) \leq \underbrace{W_2(q_{k+1}, q_{k+1}')}_{\text{Drift Term}} + \underbrace{W_2(q_{k+1}', \pi^*)}_{\text{Contraction Term}}. \tag{11}$$

The term $W_2(q_{k+1}', \pi^*)$ represents the distance to the target after one exact Langevin step. Applying Lemma A.4, we have

$$W_2(q_{k+1}', \pi^*) \leq (1 - \lambda\eta)W_2(q_k, \pi^*) + O(\eta^{1.5}), \tag{12}$$

where $\lambda$ is the strong-convexity constant of the planning potential.

The term $W_2(q_{k+1}, q'_{k+1})$ measures the divergence caused by the score approximation error. According to **Lemma A.3**, the Wasserstein distance is upper bounded by the root mean square error of our specific synchronous coupling:

$$
\begin{aligned}
W_2^2(q_{k+1}, q'_{k+1}) &\leq \mathbb{E}[\|a_{k+1} - a'_{k+1}\|_2^2] \\
&= \mathbb{E}\left[\left\|\left(a_k + \eta S(a_k) + \sqrt{2\eta}\epsilon_k\right) - \left(a_k + \eta \nabla \log \pi^*(a_k) + \sqrt{2\eta}\epsilon_k\right)\right\|_2^2\right] \\
&= \eta^2 \mathbb{E}[\|S(a_k) - \nabla \log \pi^*(a_k)\|_2^2].
\end{aligned}
$$

Using the bounded score error assumption yields

$$
W_2^2(q_{k+1}, q'_{k+1}) \leq \eta^2 \delta^2 \implies W_2(q_{k+1}, q'_{k+1}) \leq \eta \delta.
$$

Substituting the bounds back into Eq. (11):

$$
W_2(q_{k+1}, \pi^*) \leq (1 - \lambda\eta) W_2(q_k, \pi^*) + \eta\delta + O(\eta^{1.5}).
$$

Ignoring high-order terms for small $\eta$, this forms a recursive inequality $D_{k+1} \leq (1 - \lambda\eta)D_k + \eta\delta$. Unrolling this recursion for $k$ steps:

$$
W_2(q_k, \pi^*) \leq (1 - \lambda\eta)^k W_2(q_0, \pi^*) + \eta\delta \sum_{j=0}^{k-1}(1 - \lambda\eta)^j.
$$

The geometric series sums to $\frac{1-(1-\lambda\eta)^k}{\lambda\eta}$. Thus:

$$
W_2(q_k, \pi^*) \leq (1 - \lambda\eta)^k W_2(q_0, \pi^*) + \frac{\delta}{\lambda}\left(1 - (1 - \lambda\eta)^k\right).
$$

Taking the limit as $k \to \infty$, since $|1 - \lambda\eta| < 1$, the first term vanishes and the series converges, yielding the asymptotic bound:

$$
\limsup_{k\to\infty} W_2(q_k, \pi^*) \leq \frac{\delta}{\lambda}.
$$

This concludes the proof. $\qquad\square$

### A.3. Proof of Theorem 4.2

We denote the true dynamics model, reward model, and value function as $f$, $R$, and $Q$, and their learned counterparts as $\hat{f}$, $\hat{R}$, and $\hat{Q}$, respectively. Given an initial latent state $z_t$, the predicted state $\hat{z}_{t+h}$ is obtained by rolling out the learned dynamics $\hat{f}$ for $h$ steps. To facilitate the theoretical analysis, we first introduce the following assumptions.

**Assumption A.5** (Bounded model Jacobians and discounted stability)**.** We assume that $\|\nabla_z \hat{f}(z, a)\| < F_z$ and $\|\nabla_a \hat{f}(z, a)\| < F_a$, and define $F = \max(F_z, F_a)$. For the model rollout $\hat{z}_{t+h+1} = \hat{f}(\hat{z}_{t+h}, a_{t+h})$ and $a_{t+h} = \text{sg}(\pi(z_{t+h}))$, let

$$
\hat{J}_h \triangleq \left\|\frac{\partial \hat{z}_{t+h}}{\partial a_t}\right\|. \tag{13}
$$

By the chain rule, we have

$$
\hat{J}_h \leq \prod_{k=1}^{h-1} \left\|\nabla_z \hat{f}(\hat{z}_{t+k}, a_{t+k})\right\| \cdot \left\|\nabla_a \hat{f}(\hat{z}_t, a_t)\right\| < F^h. \tag{14}
$$

Besides, we assume $0 < \gamma F < 1$, where $\gamma$ is the discount factor.

**Assumption A.6** (L-Smoothness of reward and value)**.** For any $(z, \hat{z}, a)$, we assume

$$
\|\nabla_z R(\hat{z}, a) - \nabla_z R(z, a)\| \leq L_R \|\hat{z} - z\|, \qquad \|\nabla_z Q(\hat{z}, a) - \nabla_z Q(z, a)\| \leq L_Q \|\hat{z} - z\|,
$$

where $L_R$ and $L_Q$ are the Lipschitz constants.

**Assumption A.7** (Gradient approximation errors). We assume the following uniform approximation error bounds hold:

$$\|\nabla_z\hat{Q} - \nabla_z Q\| \leq E_{Qz}, \quad \|\nabla_a\hat{Q} - \nabla_a Q\| \leq E_{Qa}, \quad \|\nabla_a\hat{R} - \nabla_a R\| \leq E_{Ra}, \quad \|\nabla_z\hat{R} - \nabla_z R\| \leq E_{Rz}. \tag{15}$$

Let $E_R = \max(E_{Ra}, E_{Rz})$ and $E_Q = \max(E_{Qa}, E_{Qz})$. In continuous control settings, the reward function is governed by stationary physical laws and learned via supervised regression, making it significantly easier to approximate than the value function, which suffers from non-stationarity and bootstrap error accumulation. Consequently, we assume $E_R \ll E_Q$. Specifically, for our convergence analysis, it suffices to require: $E_R < (1 - \gamma F) E_Q$.

**Assumption A.8** (Eventual rollout accuracy). Fix a planning horizon $H$. For any $\delta > 0$, there exists a training step $i_0(\delta, H)$ such that for all $i \geq i_0$, along trajectories visited by the planner,

$$\max_{1 \leq h \leq H} \|\hat{z}_{t+h} - z_{t+h}\| \leq \delta, \qquad \max_{1 \leq h \leq H} \left\| \frac{\partial \hat{z}_{t+h}}{\partial a_t} - \frac{\partial z_{t+h}}{\partial a_t} \right\| \leq \delta.$$

This assumption is justified by the fact that the transition dynamics also follow a stationary distribution. With sufficient training, the learned model can achieve a near-perfect approximation of the underlying physical laws, minimizing rollout accumulation errors.

*Proof.* Consider the true $H$-step look-ahead return and its model-based counterpart:

$$\mathcal{J}(z_t, a_t) = \sum_{h=0}^{H-1} \gamma^h R(z_{t+h}, a_{t+h}) + \gamma^H Q(z_{t+H}, a_{t+H}),$$

$$\hat{\mathcal{J}}(z_t, a_t) = \hat{R}(z_t, a_t) + \sum_{h=1}^{H-1} \gamma^h \hat{R}(\hat{z}_{t+h}, a_{t+h}) + \gamma^H \hat{Q}(\hat{z}_{t+H}, a_{t+H}), \tag{16}$$

By the chain rule, the gradient of the return with respect to $a_t$ is

$$\nabla_a \mathcal{J}(z_t, a_t) = \nabla_a R(z_t, a_t) + \sum_{h=1}^{H-1} \gamma^h (\frac{\partial z_{t+h}}{\partial a_t})^\top \nabla_z R(z_{t+h}, a_{t+h}) + \gamma^H (\frac{\partial z_{t+H}}{\partial a_t})^\top \nabla_z Q(z_{t+H}, a_{t+H}). \tag{17}$$

Accordingly, the difference between the true return gradient and the gradient induced by the learned models is

$$\nabla_a \hat{\mathcal{J}}(\hat{z}_t, a) - \nabla_a \mathcal{J}(z_t, a) = \left( \nabla_a \hat{R}(z_t, a_t) - \nabla_a R(z_t, a_t) \right)$$
$$+ \sum_{h=1}^{H-1} \gamma^h \underbrace{\left( (\frac{\partial \hat{z}_{t+h}}{\partial a_t})^\top \nabla_z \hat{R}(\hat{z}_{t+h}, a_{t+h}) - (\frac{\partial z_{t+h}}{\partial a_t})^\top \nabla_z R(z_{t+h}, a_{t+h}) \right)}_{A}$$
$$+ \gamma^H \underbrace{\left( (\frac{\partial \hat{z}_{t+H}}{\partial a_t})^\top \nabla_z \hat{Q}(\hat{z}_{t+H}, a_{t+H}) - (\frac{\partial z_{t+H}}{\partial a_t})^\top \nabla_z Q(z_{t+H}, a_{t+H}) \right)}_{B}. \tag{18}$$

Since $A$ and $B$ share the same form, we only analyze $A$, the bound for $B$ follows analogously. For $A$, we add and subtract $\left( \frac{\partial \hat{z}_{t+h}}{\partial a_t} \right)^\top \nabla_z R(\hat{z}_{t+h}, a_{t+h})$ and $\left( \frac{\partial \hat{z}_{t+h}}{\partial a_t} \right)^\top \nabla_z R(z_{t+h}, a_{t+h})$, which yields

$$A = \underbrace{(\frac{\partial \hat{z}_{t+h}}{\partial a_t})^\top \nabla_z \hat{R}(\hat{z}_{t+h}, a_{t+h}) - (\frac{\partial \hat{z}_{t+h}}{\partial a_t})^\top \nabla_z R(\hat{z}_{t+h}, a_{t+h})}_{A_1}$$
$$+ \underbrace{(\frac{\partial \hat{z}_{t+h}}{\partial a_t})^\top \nabla_z R(\hat{z}_{t+h}, a_{t+h}) - (\frac{\partial \hat{z}_{t+h}}{\partial a_t})^\top \nabla_z R(z_{t+h}, a_{t+h})}_{A_2} \tag{19}$$
$$+ \underbrace{(\frac{\partial \hat{z}_{t+h}}{\partial a_t})^\top \nabla_z R(z_{t+h}, a_{t+h}) - (\frac{\partial z_{t+h}}{\partial a_t})^\top \nabla_z R(z_{t+h}, a_{t+h})}_{A_3}.$$

Using the Assumptions A.6–A.8 and Lemma A.2, we obtain the following bounds:

$$\|A_1\| \leq \|\frac{\partial \hat{z}_{t+h}}{\partial a_t}\| \|\nabla_z \hat{R}(\hat{z}_{t+h}, a_{t+h}) - \nabla_z R(\hat{z}_{t+h}, a_{t+h})\| \leq \hat{J}_h E_R,$$

$$\|A_2\| \leq \|\frac{\partial \hat{z}_{t+h}}{\partial a_t}\| \|\nabla_z R(\hat{z}_{t+h}, a_{t+h}) - \nabla_z R(z_{t+h}, a_{t+h})\| \leq \hat{J}_h L_R \delta = O(\delta), \tag{20}$$

$$\|A_3\| \leq \|\left((\frac{\partial \hat{z}_{t+h}}{\partial a_t})^\top - (\frac{\partial z_{t+h}}{\partial a_t})^\top\right)\| \|\nabla_z R(z_{t+h}, a_{t+h})\| \leq \delta \|\nabla_z R(z_{t+h}, a_{t+h})\| = O(\delta).$$

Since $\delta > 0$ can be chosen arbitrarily small by Assumption. A.8, the residual terms proportional to $\delta$ can be made negligible. Thus, focusing on the dominant approximation terms gives

$$\|A\| \leq \hat{J}_h E_R, \quad \|B\| \leq \hat{J}_H E_Q. \tag{21}$$

Therefore, we have

$$
\begin{aligned}
\sup \left\| \nabla_a \hat{\mathcal{J}}(\hat{z}_t, a_t) - \nabla_a \mathcal{J}(z_t, a_t) \right\| &\leq E_R + \sum_{h=1}^{H-1} \gamma^h \hat{J}_h E_R + \gamma^H \hat{J}_H E_Q \\
&< E_R \sum_{h=0}^{H-1} (\gamma F)^h + (\gamma F)^H E_Q \\
&= E_R \frac{1 - (\gamma F)^H}{1 - \gamma F} + (\gamma F)^H E_Q \\
&< E_Q
\end{aligned}
\tag{22}
$$

The last inequality follows directly from the condition $E_R < (1 - \gamma F) E_Q$ in Assumption A.7. Consequently, we establish the final conclusion:

$$\sup \left\| \nabla_a \hat{\mathcal{J}}(z_t, a) - \nabla_a \mathcal{J}(z_t, a) \right\| < \sup \left\| \nabla_a \hat{Q}(z_t, a) - \nabla_a Q(z_t, a) \right\| \tag{23}$$

This inequality demonstrates that the gradient derived from the unrolled world model exhibits strictly lower estimation error compared to the direct gradient of the value function. In the context of our LaRO, this superior gradient fidelity ensures that the sampling dynamics are guided by more accurate gradients, thereby reducing the risk of mode collapse or divergence caused by value approximation artifacts. This concludes the proof.

$\square$

### A.4. Important Remarks

*Remark* A.9 (On the Optimization Scope). We distinguish the optimization scope between the two planner components. In MPPI, the look-ahead return $\mathcal{J}(s_t, a_{t:t+H-1})$ is evaluated over the full action sequence $a_{t:t+H-1}$. In contrast, LaRO defines the objective $\mathcal{J}(s_t, a_t)$ strictly with respect to the immediate action $a_t$, where subsequent actions are generated by the RL policy $\pi$. This is motivated by the Receding Horizon Control principle: since only the first action is executed, and the gradient of the cumulative return can backpropagate through the dynamics and policy to $a_t$, optimizing $a_t$ directly is sufficient and computationally more efficient than optimizing the entire high-dimensional sequence.

*Remark* A.10 (On the Convexity Assumption). While the strict assumption of global strong convexity in Theorem 4.1 is employed to derive explicit contraction rates in the Wasserstein metric, in practice, the planning landscape is often non-convex and multimodal. In such cases, Langevin dynamics is theoretically known to traverse between modes over sufficiently long timescales (Raginsky et al., 2017). Our analysis effectively characterizes the rapid local convergence properties within the basin of attraction of a dominant mode, which complements the global exploration capability provided by the injected noise.

# B. Environmental Details

## B.1. Toy Experiments

The planning objective function in our toy experiments is

$$\mathcal{J}_{\text{plan}}(\mathbf{x}) = \exp\left(-\frac{1}{2}\mathbf{x}^T\Sigma^{-1}\mathbf{x}\right) + 0.045\sin(1.0x + 0.6y)\cos(0.8x - 0.4y)$$

where $\mathbf{x} = [x, y]^T$, and $\Sigma$ is given by

$$\Sigma = \begin{bmatrix} \cos\frac{\pi}{6} & -\sin\frac{\pi}{6} \\ \sin\frac{\pi}{6} & \cos\frac{\pi}{6} \end{bmatrix} \begin{bmatrix} 1 & 0 \\ 0 & 0.09 \end{bmatrix} \begin{bmatrix} \cos\frac{\pi}{6} & -\sin\frac{\pi}{6} \\ \sin\frac{\pi}{6} & \cos\frac{\pi}{6} \end{bmatrix}^\top.$$

## B.2. Benchmark Introduction

**DeepMind Control Suite.** The evaluation suite consists of the 7 most challenging control tasks for the *dog* and *humanoid* domains, divided into: (1) *Standing tasks*, requiring the preservation of upright equilibrium, and (2) *Moving tasks*, which entail the additional objective of reaching a target velocity. For tasks involving movement, we compute the total reward by scaling the posture reward with the speed tracking reward: ***Reward*** = (***Standing reward***) × (***Forward velocity reward***).

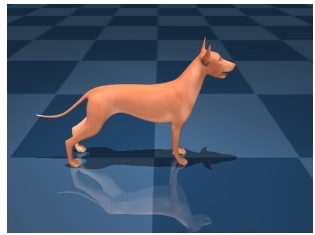

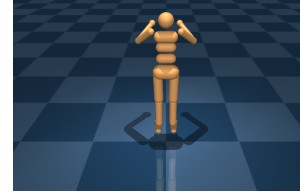

***Standing reward***: Encourages the agent to maintain an upright posture.

***Forward velocity reward***: Ensures the agent moves at the target speed ($1\,\text{m/s}$ for dog-walk, $3\,\text{m/s}$ for dog-trot, $9\,\text{m/s}$ for dog-run, $1\,\text{m/s}$ for humanoid-walk and $10\,\text{m/s}$ for humanoid-run).

*Figure 4.* Dog    *Figure 5.* Humanoid

**Humanoid Bench**. We consider 7 typical locomotion tasks involving a Unitree H1hand robot. This robot is initialized to a standing position, with random noise added to all joint positions during each episode reset. Their specific goals are presented below.

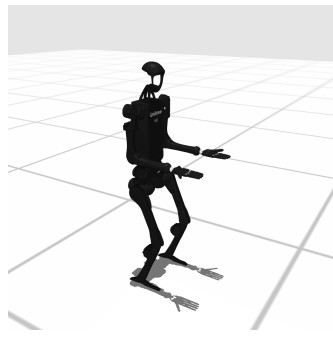

***Objective.*** Maintain a standing pose.

***Reward***: $R(s, a) = \texttt{stable} \times (0.5 \times \texttt{still}_x + 0.5 \times \texttt{still}_y)$, where the $\texttt{still}$ terms penalize non-zero velocities to encourage stationary balance. $\texttt{stable}$ favors maintaining a stable and energy-efficient standing status.

***Termination.*** 1000 steps, or when $z_{\text{pelvis}} < 0.2$.

*Figure 6.* Stand

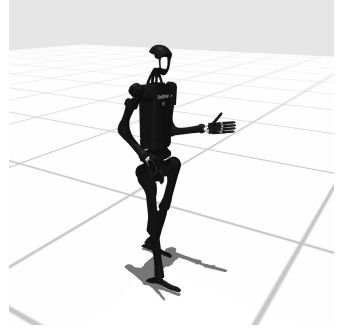

*Figure 7.* Walk

***Objective.*** Keep forward velocity close to $1\,\mathrm{m/s}$ without falling to the ground.

***Reward***: $R(s,a) = \mathtt{stable} \times \mathtt{tol}(v_x, (1, \infty), 1)$, where $\mathtt{tol}$ encourages the agent to maintain a forward velocity $v_x$ above $1\,\mathrm{m/s}$, thereby promoting low-speed locomotion.

***Termination.*** 1000 steps, or when $z_{\mathrm{pelvis}} < 0.2$.

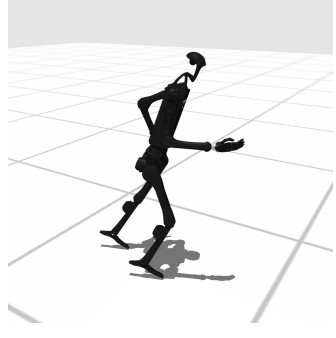

*Figure 8.* Run

***Objective.*** Keep forward velocity close to $5\,\mathrm{m/s}$ without falling to the ground.

***Reward***: $R(s,a) = \mathtt{stable} \times \mathtt{tol}(v_x, (5, \infty), 5)$, where $\mathtt{tol}$ encourages the agent to maintain a forward velocity $v_x$ above $5\,\mathrm{m/s}$, thereby promoting high-speed locomotion.

***Termination.*** 1000 steps, or when $z_{\mathrm{pelvis}} < 0.2$.

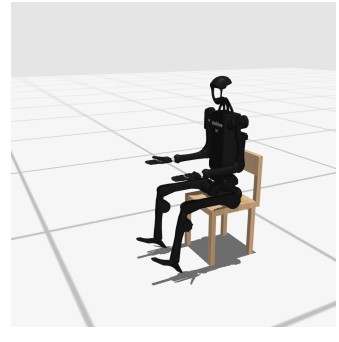

*Figure 9.* Sit

***Objective.*** Sit onto a chair situated closely behind.

***Reward***: $R(s,a) = (0.5 \cdot \mathtt{sitting\_z} + 0.5 \cdot \mathtt{sitting\_x} \cdot \mathtt{sitting\_y}) \times \mathtt{upright} \times \mathtt{posture} \times e \times \mathtt{mean}(\mathtt{still\_x}, \mathtt{still\_y})$, where $e$ is an energy penalty term, $\mathtt{sitting\_x}$, $\mathtt{sitting\_y}$, and $\mathtt{sitting\_z}$ measure the robot's positional tolerance relative to the chair.

***Termination.*** 1000 steps, or when $z_{\mathrm{pelvis}} < 0.5$.

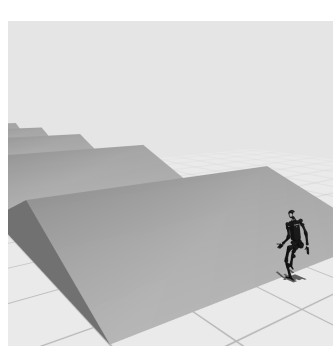

*Figure 10.* Slide

***Objective.*** Walk over an iterating sequence of upward and downward slides at $1\,\mathrm{m/s}$.

***Reward***: $R(s,a) = e \times \mathtt{tol}(v_x, (1, +\infty), 1) \times \mathtt{upright} \times (\mathtt{foot\_left} \times \mathtt{foot\_right})$, where $\mathtt{foot\_left}$ and $\mathtt{foot\_right}$ measure the vertical distance between the head and left/right foot respectively, ensuring proper foot positioning.

***Termination.*** 1000 steps, or when $z_{\mathrm{proj}} < 0.6$.

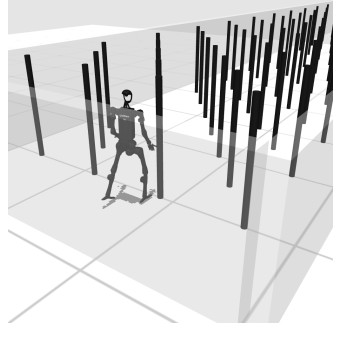

***Objective.*** Travel forward over a dense forest of high thin poles, without colliding with them.

***Reward***: $R(s, a) = \gamma_{\text{collision}} \times \big(0.5 \times \text{stable} + 0.5 \times \texttt{tol}(v_x, (1, +\infty), 1)\big)$, where the collision penalty $\gamma_{\text{collision}}$ equals 0.1 if the robot collides with a pole, and 1 otherwise.

***Termination.*** 1000 steps, or when $z_{\text{pelvis}} < 0.6$.

*Figure 11.* Pole

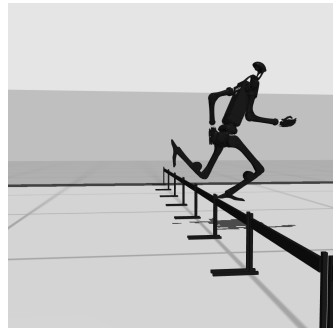

***Objective.*** Keep forward velocity close to $5\,\text{m/s}$ without falling to the ground.

***Reward***: $R(s, a) = \text{stable} \times \texttt{tol}(v_x, (5, \infty), 5) \times \gamma_{\text{collision}}$, which penalizes colliding with hurdle.

***Termination.*** 1000 steps.

*Figure 12.* Hurdle

### B.3. Reproducibility Statement & Detailed Hyperparameters

We base all our experiments on the released official BOOM codebase https://github.com/molumitu/BOOM_MBRL. We adopt their hyperparameter settings without additional tuning and use the same configuration across all previously demonstrated tasks. The details are listed in Table 4.

In this paper, we evaluate each algorithm for each tasks over five random seeds. The CPU used is the AMD EPYC 9K84 16-core Processor, and the GPU used is NVIDIA H20. Taking *H1hand-run* task in the Humanoid Bench as an example, the time taken to train 1M environment steps is around 25 hours.

For SAC and DreamerV3, we report the baseline results released by TD-MPC2, which are obtained from the official repositories of https://github.com/denisyarats/pytorch_sac and https://github.com/danijar/dreamerv3, respectively. For BMPC, we use the official implementation https://github.com/wertyuilife2/bmpc and align the settings such as environment interaction steps and evaluation protocol for a fair comparison.

## C. Extra Experiments

### C.1. Adaptive Hybrid Planning

While our MLAP framework runs MPPI and LaRO in parallel at every step, a natural question is whether LaRO can be invoked more selectively to reduce inference overhead without sacrificing performance. To this end, we preliminarily implement an **Adaptive Hybrid Planning (AHP)** variant that triggers LaRO only when MPPI appears to plateau, and study two triggering criteria:

- **AHP-I (improvement-based):** Activates LaRO when the per-iteration improvement in MPPI's weighted return falls below a fixed threshold, indicating that zero-order sampling has effectively stalled.

- **AHP-V (variance-based):** Activates LaRO when the sampling variance of MPPI's action distribution contracts below a threshold, suggesting that the search has collapsed into a narrow local basin where gradient guidance is most needed.

*Table 4.* Hyperparameter settings.

| Hyperparameter | Value | | Hyperparameter | Value |
|---|---|---|---|---|
| **Training** | | | | |
| Learning rate | $3 \times 10^{-4}$ | | Target network update rate | 0.5 |
| Encoder learning rate | $1 \times 10^{-4}$ | | Discount factor ($\gamma$) | 0.99 |
| Sample batch size | 1 | | Gradient Clipping Norm | 20 |
| Replay batch size | 256 | | Optimizer | Adam |
| Buffer size | $1 \times 10^{6}$ | | Loss norm | Moving $(5\%, 95\%)$ |
| Steps | $1 \times 10^{6}$ | | Sampling | Uniform |
| **Score-Augmented World Model** | | | | |
| Reward loss coefficient ($c_r$) | 0.1 | | Dynamics loss coefficient ($c_f$) | 20 |
| Value loss coefficient ($c_q$) | 0.1 | | score loss coefficient ($c_s$) | 1.0 |
| Number of value bins | 101 | | Value dropout rate | 1% |
| Warmup steps | 1000 | | Value functions ensemble | 5 |
| **Planner (MLAP)** | | | | |
| MPPI Iterations | 6 (8 if $\|\mathcal{A}\| > 20$) | | Minimum planner std | 0.05 |
| LaRO Iterations | 5 | | Maximum planner std | 2 |
| LaRO step size | $5 \times 10^{-3}$ | | Horizon | 3 |
| Population size | 512 | | Policy prior samples | 24 |
| Number of elites | 64 | | Temperature parameter ($\alpha$) | 1.0 |
| **Actor** | | | | |
| Minimum policy log std | -10 | | Entropy coefficient ($\alpha$) | $1 \times 10^{-4}$ |
| Maximum policy log std | 2 | | | |
| **Architecture (around 5M parameters in total)** | | | | |
| Encoder layers | 2 | | Latent space dimension | 512 |
| Encoder dimension | 256 | | Task embedding dimension | 96 |
| MLP hidden layer dimension | 512 | | Q function drop out rate | 0.01 |
| MLP activation | Mish | | MLP Normalization | LayerNorm |

We evaluate both strategies on the H1hand-run task and report average TAR and per-step inference latency in Table 5. Both AHP variants substantially outperform the MPPI-only baseline (BOOM: 329.3) while reducing inference time relative to the always-on MLAP. AHP-I retains most of the performance gain (801.5 vs. 823.0) at a slightly lower latency (32.38 ms vs. 34.93 ms), whereas AHP-V reduces latency further (31.46 ms) but with a larger performance gap. These results suggest that adaptive triggering is a promising direction for improving real-time efficiency, though always-on MLAP remains the stronger choice when peak performance is the priority.

*Table 5.* Comparison of adaptive triggering strategies on H1hand-run (averaged over 5 seeds). BOOM (MPPI-only) serves as the baseline.

| Planner | TAR | Inference Time (ms) |
|---|---|---|
| MLAP (Ours) | **823.0** $\pm 48.8$ | 34.93 |
| AHP-I | $801.5 \pm 24.7$ | 32.38 |
| AHP-V | $783.7 \pm 32.8$ | 31.46 |
| MPPI-only (BOOM) | $329.3 \pm 38.0$ | 28.87 |

## C.2. Expanded Benchmark Experiments

To validate the generality of BOOM-L beyond the 14 primary tasks reported in the main paper, we expand the evaluation to an additional 21 tasks across three supplementary suites, each evaluated over 5 seeds against the two strongest baselines, BMPC and BOOM. The three suites are: **DMC-Easy** (7 tasks from the Walker, Quadruped, Hopper, and Cheetah domains), **Extra H-Bench with Hand** (7 additional H1hand tasks including manipulation-oriented challenges such as spoon, maze, and bookshelf), and **H-Bench without Hand** (7 tasks on the standard H1 robot without dexterous hands).

Training curves are shown in Figures 13, 14, and 15, and average TAR for each suite is summarized in Table 6.

*Table 6.* Average TAR on three supplementary benchmark suites (5 seeds; bold = best).

| Benchmark (AVG.) | BMPC | BOOM | BOOM-L |
|---|---|---|---|
| Extra H-Bench (Hand) | $367.4 \pm 79.3$ | $367.8 \pm 34.1$ | $\mathbf{506.3} \pm 37.7$ |
| DMC Suite (Easy) | $861.3 \pm 7.0$ | $867.3 \pm 10.0$ | $\mathbf{880.6} \pm 9.8$ |
| H-Bench (w/o Hand) | $460.6 \pm 89.7$ | $737.6 \pm 30.8$ | $\mathbf{775.5} \pm 7.1$ |

BOOM-L achieves the best average performance across all three suites. The gains are most pronounced on the challenging Extra H-Bench (Hand) tasks (+37.6% over BOOM), where complex coordination and long-horizon objectives create the sharp, rugged return landscapes that most benefit from gradient guidance. On DMC-Easy, where all planning-based methods are already close to the performance ceiling, BOOM-L still achieves a consistent edge, confirming that gradient-guided refinement does not degrade performance on simpler tasks. Taken together with the main 14-task results, BOOM-L consistently outperforms strong baselines across 35 diverse continuous-control tasks, providing strong evidence for its generality.

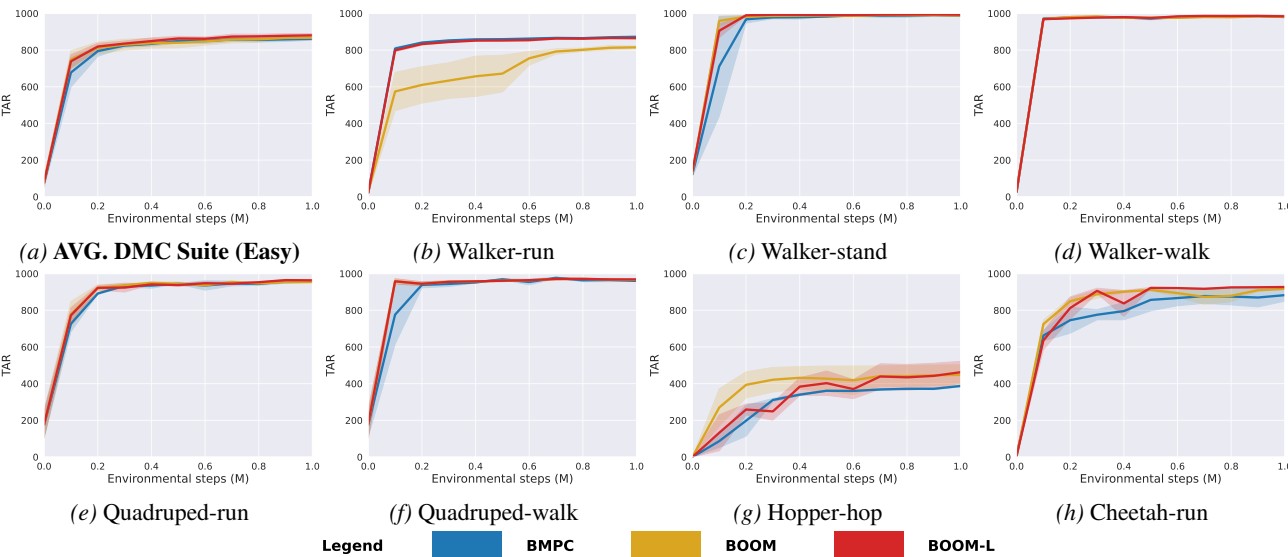

*Figure 13.* **Training curves on DMC Suite (Easy).** The solid lines represent the mean, while the shaded regions indicate the confidence interval over five runs. The average performance curves for the two benchmarks appear at the left corner of the 1st and 3rd rows, respectively, highlighted in **bold**.

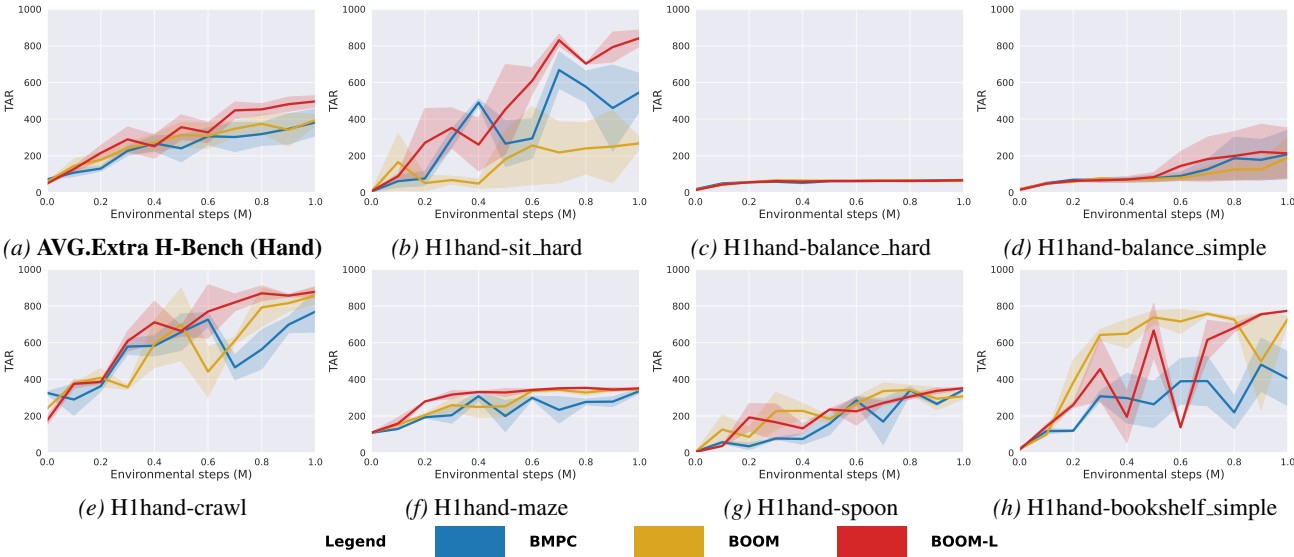

*Figure 14.* **Training curves on Extra H-Bench (Hand).** The solid lines represent the mean, while the shaded regions indicate the confidence interval over five runs. The average performance curves for the two benchmarks appear at the left corner of the 1st and 3rd rows, respectively, highlighted in **bold**.

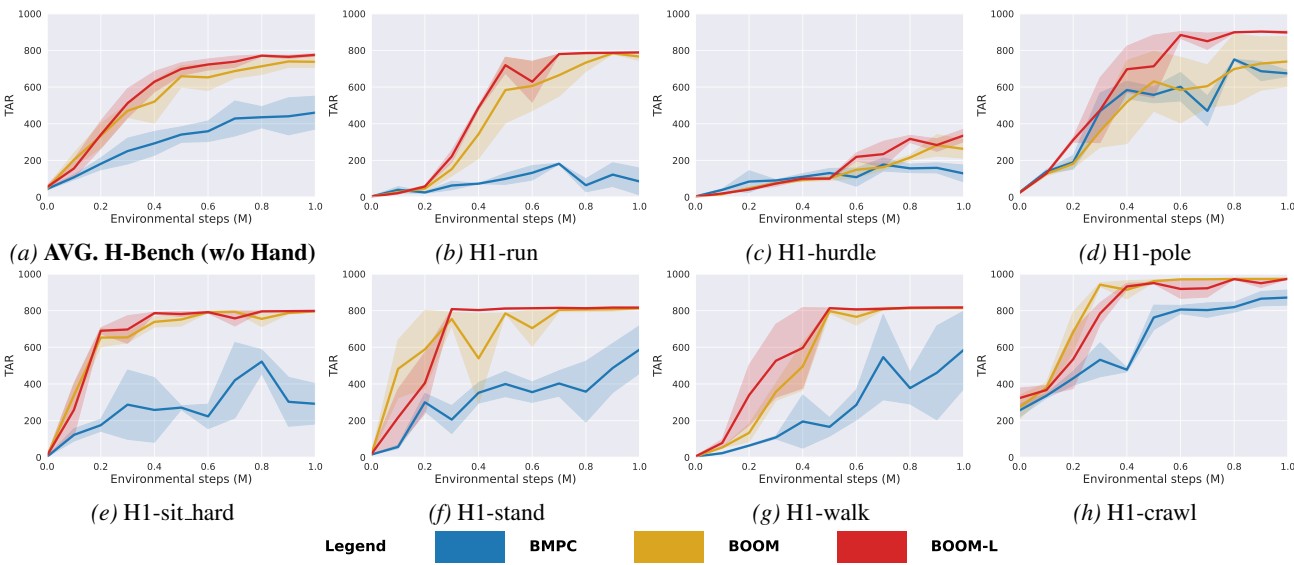

*Figure 15.* **Training curves on H-Bench (w/o Hand).** The solid lines represent the mean, while the shaded regions indicate the confidence interval over five runs. The average performance curves for the two benchmarks appear at the left corner of the 1st and 3rd rows, respectively, highlighted in **bold**.

