# OpenReview forum: "Langevin Rollout Optimization for Modelic Reinforcement Learning"
_ICML.cc/2026/Conference — ICML 2026 regular_

### Official Review · Reviewer_hpzg · 2026-03-09

**Soundness:** 4
**Presentation:** 3
**Significance:** 4
**Originality:** 4
**Overall Recommendation:** 5
**Confidence:** 4

**Summary:**

This paper presents a reinforcement learning algorithm, "Langevin Rollout Optimization (LaRO)". TD-MPC and BOOM used Model Predictive Path Integral (MPPI) based on zero-degree MPC optimization. Rather than MPPI, the authors proposed a planning algorithm that samples actions from the Boltzmann distribution. Also, to avoid BPTT, they used a score network. Lastly, they paralled MPPI and LaRO to execute the candidate with higher look-ahead returns.

**Compliance With Llm Reviewing Policy:**

Affirmed.

**Key Questions For Authors:**

See my comments within Weaknesses.

**Limitations:**

See my comments within Weaknesses.

**Strengths And Weaknesses:**

Strengths:\
$\bullet$ Proposed algorithm is creative and intuitive. \
$\bullet$ Mathematical progress for optimization and substitution of the score network is clever. \
$\bullet$ The presented performance of BOOM-L is attractive and justifies their contribution.

Weakness:\
$\bullet$ Authors suggest Theorem 4.2. I wonder if this Theorem is about 'Model-based return gradients > Value Gradients (With Langevin Rollout Optimization)' or 'Model-based return gradients > Value Gradients (Every circumstance)'. \
$\bullet$ The authors assumed that MMPI's optimization efficiency was insufficient compared to LaRO. But actually, in their Ablation study, it was not. LaRO is similar to MPPI, and only the two-way can be improved, not the one-way.\
$\bullet$ In my opinion, LaRo can not evade the point of the weakness in the statistical credentials and selective experiment at this moment. I wonder why authors just train not the whole HumanoidBench (Hand). Furthermore, there are only 3 seeds; this is not fair, especially since the suggested algorithms are based on distribution. I strongly recommend authors to experiment with Gym, DMC-Easy, DMC-Hard, and HBench (without Hand, with Hand) using at least 5 or 10 seeds if they want to claim 'establishing a new state-of-the-art'. The current declaration is unacceptable. Also, there are h1-sit-simple and h1-sit-hard in HBench, but the authors just write H1hand-sit. I hope to reveal a precise Suite of experiments. \
$\bullet$ In the paper, there are no limitations of the research.

---

> ### Author Rebuttal · Authors · 2026-03-31
>
> We sincerely appreciate your constructive feedback. Below are our responses to your comments.
>
> # W1: Generality of Theorem 4.2
> We clarify that Theorem 4.2 is intended as a more general statement than one restricted to LaRO, but not as an unconditional every-circumstance claim. Rather, it is a conditional result: under Assumptions A.5-A.8, gradients computed by unrolling the dynamics admit a tighter error bound than gradients derived directly from a bootstrapped value network.
>
> # W2: LaRO vs. MPPI Standalone Performance
> We clarify that our claim is not that pure LaRO universally outperforms pure MPPI. Rather, our core argument is that purely zero-order planners (like MPPI) discard valuable first-order gradient information, failing to fully exploit the world model’s potential. Our core contribution is introducing LaRO to recover this missing local precision, alongside a synergistic MLAP framework that seamlessly combines MPPI’s global coverage with LaRO’s local gradient precision. Naturally, because pure LaRO lacks MPPI's global exploration, it is not guaranteed to strictly outperform MPPI on its own (as shown in our ablation study).
>
> # W3: Expanded Experiments
> We appreciate this constructive feedback and agree that our original 14-task evaluation was too narrow to support a broad SOTA claim. Initially, these tasks were selected for their nature as high-dimensional control problems, especially in H-Bench (Hand), where we focused on relatively difficult tasks like H1hand-run.
>
> To address your concerns, we have expanded our evaluation suite into a broader benchmark encompassing **35 tasks**, all run with **5 seeds**:
> (i) **Original 14 tasks (DMC-Hard & H-Bench (Hand))**: We re-evaluated all algorithms with 5 seeds. The results are shown in [Figure 6](https://anonymous.4open.science/r/ICML26_BOOML_rebuttal/Figure6-main_exps_5seed.pdf) and [Table 1](https://anonymous.4open.science/r/ICML26_BOOML_rebuttal/Table1-main_exps_5seed.pdf). We also corrected the naming typo from `H1hand-sit` to `H1hand-sit_simple` to clarify the precise experiment suite.
> (ii) **Extra H-Bench (Hand)**
> (iii) **DMC-Easy**
> (iv) **H-Bench (without Hand)**
> Due to the limited rebuttal window, for the newly added suites (ii)-(iv), we evaluated **7 tasks per suite** and focused our 5-seed runs on the strongest prior methods: BMPC and BOOM. The results are shown in [Figures 7](https://anonymous.4open.science/r/ICML26_BOOML_rebuttal/Figure7-extra_hbench_hand.pdf), [8](https://anonymous.4open.science/r/ICML26_BOOML_rebuttal/Figure8-dmc_easy.pdf), [9](https://anonymous.4open.science/r/ICML26_BOOML_rebuttal/Figure9-hbench_nohand.pdf) and summarized in the table below, which reports the average TAR for each suite. Across all suites, BOOM-L consistently achieves the best performance, including slight gains in simpler environments such as DMC-Easy, where all methods are already close to the performance ceiling.
>
> | Benchmark (AVG. ) | BMPC | BOOM | BOOM-L |
> | - | - | - | - |
> | Extra H-Bench (Hand) | 367.4 $\pm$ 79.3 | 367.8 $\pm$ 34.1 | **506.3** $\pm$ 37.7 |
> | DMC Suite (Easy) | 861.3 $\pm$ 7.0 | 867.3 $\pm$ 10.0 | **880.6** $\pm$ 9.8 |
> | H-Bench (w/o Hand) | 460.6 $\pm$ 89.7 | 737.6 $\pm$ 30.8 | **775.5** $\pm$ 7.1 |
>
> We regret that we could not add **Gym** now because the expanded 35 tasks, 5-seed evaluation already required over 450 GPU-days (BMPC, BOOM, and BOOM-L all take ~1 day/trial), exhausting our compute budget in the rebuttal window. We therefore prioritized DMC and H-Bench, which are standard continuous-control benchmarks in modern MBRL literature (e.g., BMPC, BOOM). We acknowledge that the current omission of Gym limits a universal SOTA claim. However, consistently outperforming the strongest baselines across 35 diverse tasks provides strong evidence that BOOM-L establishes a new SOTA **among planning-based MBRL methods**. We plan to include the Gym evaluations and remaining baseline runs in a later version.
>
> # W4: Limitations
> Thank you for pointing out this omission. We will add a "Limitations" section in the revised manuscript.
>
> A primary limitation of our method is the trade-off between planning precision and computational efficiency. While our MLAP framework significantly boosts performance on highly challenging tasks by running MPPI and LaRO in parallel, it naturally incurs a slightly higher inference latency than purely zero-order planners. To mitigate this, our future work will focus on developing adaptive, lightweight action-selection mechanisms. By dynamically triggering gradient-guided refinement only when necessary, we aim to further optimize real-time computational efficiency without sacrificing control performance.
>
> # Final thanks
> Thank you again for your time, effort, and professionalism. We hope our responses address your concerns. We are happy to provide additional details if needed, and we look forward to further discussion.

---

> > ### Author Rebuttal · Reviewer_hpzg · 2026-04-01
> >
> > I thank the authors for their detailed response and the additional experiments. My concerns regarding the statistical significance of the results have been largely addressed. I hope the final version or the camera-ready manuscript will include the results for the Gym environments and the rest of the suggested benchmarks. Accordingly, I will raise my score to reflect these improvements.

---

> > > ### Author Response · Authors · 2026-04-01
> > >
> > > Thank you very much for your thoughtful follow-up and positive feedback. We will include the comprehensive evaluations on the Gym environments, along with the rest of your suggested benchmarks, in the final camera-ready version of the manuscript. Thank you again for your highly constructive critiques, which have significantly strengthened the rigor of our paper.

---

### Official Review · Reviewer_8cYq · 2026-03-13

**Soundness:** 3
**Presentation:** 3
**Significance:** 3
**Originality:** 4
**Overall Recommendation:** 5
**Confidence:** 3

**Summary:**

This paper introduces **BOOM-L**, a model-based reinforcement learning algorithm that enhances online planning by integrating first-order gradient information via Langevin dynamics (**LaRO**). To address the computational bottleneck of back-propagation through time (BPTT) during real-time inference, the authors propose a score-augmented world model that amortizes return gradients into a latent score network. Furthermore, the method employs a Maximum Look-Ahead Planning (**MLAP**) framework to balance local, gradient-guided precision with global, sampling-based exploration. Extensive evaluations across 14 high-dimensional continuous control benchmarks, including the complex Humanoid Bench, demonstrate that BOOM-L achieves state-of-the-art sample efficiency and asymptotic performance, significantly outperforming traditional zero-order sampling baselines in navigating rugged reward landscapes.

**Compliance With Llm Reviewing Policy:**

Affirmed.

**Final Justification:**

Most of my concerns has been addressed. I am especially glad to see my Q3 is beneficial and answered fully by the new set of experiments.

**Key Questions For Authors:**

- See weaknesses 1.
- See weaknesses 2.
- This might be too tricky. Have you experimented with a more adaptive, less computationally redundant approach for MLAP, such as dynamically triggering LaRO only when MPPI reaches a performance plateau, rather than running both in parallel and picking the max?

**Limitations:**

Yes

**Strengths And Weaknesses:**

**Strengths**
- The paper introduces Langevin dynamics (LaRO) to MBRL, successfully shifting the online planner from relying on zero-order sampling (MPPI) to utilizing gradient-guided optimization for precise action refinement. This is novel.
- The Score-Augmented World Model elegantly distills return gradients into a latent score network. This completely bypasses the prohibitive computational cost of BPTT, achieving a $>7\times$ speedup and making real-time gradient planning feasible.
- BOOM-L achieves SOTA results across 14 demanding continuous control benchmarks. The overall empirical results looks good.

**Weaknesses**
- While inference is fast, training the Score Network requires expensive BPTT to generate target gradients. The paper lacks an analysis of this added wall-clock training cost compared to baselines。
- LaRO introduces new, potentially brittle hyperparameters (Langevin step size, noise scale, optimization steps). Deeper ablation studies on their robustness are needed.
- Gradient-guided search is highly sensitive to model inaccuracies in out-of-distribution (OOD) states, potentially leading to worse exploitation errors than zero-order sampling.

---

> ### Author Rebuttal · Authors · 2026-03-31
>
> We sincerely appreciate your constructive feedback. Below are our responses to your comments.
>
> # W1 & Q1: Wall-clock Training Cost
> We agree that the single-task efficiency analysis in **Appendix B.4** was limited. We therefore expand the 1M-step wall-clock evaluation to four tasks (two from DMC and two from H-Bench):
> | Training Time (h) | Dog-run | Humanoid-run | H1hand-slide | H1hand-run | Average |
> | - | - | - | - | - | - |
> | BOOM | 20.27  | 18.39  | 23.16  | 19.07  | 20.22  |
> | BOOM-L | 25.08  | 23.74  | 28.29  | 24.91  | 25.51  |
> | BOOM-L w/o Score | 42.53  | 38.61  | 46.86  | 40.34  | 42.09 |
>
> We will also add detailed analysis in the revision:
> (i) Why BOOM-L is faster than BOOM-L w/o score: BPTT-based LaRO requires **$K$ BPTT** passes per environment step ($K=5$: Langevin iterations). Score-based LaRO requires **zero** online BPTT, shifting expensive BPTT entirely to the network update phase, where it is computed only **once** per batch.
> (ii) Why BOOM-L is slower than BOOM: The overhead stems from MLAP's action selection and the BPTT computations required to train the score network.
>
> # W2 & Q2: Hyperparameter robustness
>
> Our original submission provided ablation studies on **noise scale** (temperature $\alpha$) and **optimization steps** (Langevin steps) for the H1hand-run task in **Figure 3 (b, c)**. Following your suggestion, we now add an ablation on **Langevin step size** ($\eta$) for the same task, shown in [Figure 2](https://anonymous.4open.science/r/ICML26_BOOML_rebuttal/Figure2-eta_ablation.pdf). The numerical results for all **three key hyperparameters** are:
>
> |  $\alpha$ | 1 (Ours) | 1e-2 | 1e-4 |
> | - | - | - | - |
> | TAR |**806.5** $\pm$ 45.7 | 669.4 $\pm$ 16.7 | 583.0 $\pm$ 2.9 |
>
> | Steps | 3 | 5 (Ours) | 8 |
> | - | - | - | - |
> | TAR | 609.6 $\pm$ 8.8 | **806.5** $\pm$ 45.7 | 735.4 $\pm$ 9.1 |
>
> | $\eta$ | 1e-3 | 5e-3 (Ours) | 1e-2 |
> | - | - | - | - |
> | TAR | 771.8 $\pm$ 38.2 | **806.5** $\pm$ 45.7 | 754.5 $\pm$ 22.7 |
>
> The results confirm BOOM-L's robustness across a reasonable range of hyperparameters, with all variants significantly outperforming the MPPI baseline (BOOM, 326.0). As discussed in our paper, the performance variations with respect to $\alpha$ and $K$ align with our theoretical design.
>
> For a **deeper cross-task analysis**, we evaluated these hyperparameters on the DMC's Humanoid-run task. As shown in [Figure 3](https://anonymous.4open.science/r/ICML26_BOOML_rebuttal/Figure3-ab_human_run.pdf), the results exhibit a similarly stable pattern.
>
> # W3: OOD sensitivity
>
> We agree that OOD sensitivity is a critical challenge. While our method does not eliminate this risk, it mitigates it through three safeguards:
> (i) **Local scope**: LaRO refines only the first executed action, reducing exposure to long-horizon compounding error;
> (ii) **Policy-prior initialization**: LaRO starts from the policy prior instead of arbitrary actions, keeping updates near visited (in-distribution) regions;
> (iii) **Stochastic & clipped updates**: Injected Langevin noise preserves exploration to prevent collapsing into spurious OOD peaks, while gradient clipping restricts unstable updates.
>
> Ultimately, OOD sensitivity is a shared MBRL challenge. However, [Figure 4](https://anonymous.4open.science/r/ICML26_BOOML_rebuttal/Figure4-consistency_loss.pdf) shows our dynamics loss converges rapidly, ensuring high model accuracy and minimizing OOD encounters for reliable gradient guidance. Future work can further alleviate this issue via early world model pre-training.
>
> # Q3: Adaptive Triggering
>
> Thank you for this insightful suggestion. To address this, we implement an **Adaptive Hybrid Planning (AHP)** variant that triggers LaRO only when MPPI appears to plateau. We consider two criteria:
>
> - **Improvement-based trigger (AHP-I)**: Activates LaRO when MPPI return improvement falls below a threshold.
> - **Variance-based trigger (AHP-V)**: Activates LaRO when MPPI sampling variance contracts below a threshold (indicating a narrow local basin).
>
> We evaluated both strategies on the H1hand-run task. The results are shown below and in [Figure 5](https://anonymous.4open.science/r/ICML26_BOOML_rebuttal/Figure5-ahp_ablation.pdf).
>
> | Planner | TAR | Inference Time |
> | - | - | - |
> | MLAP (Ours) | **806.5** $\pm$ 45.7 | 34.93 ms |
> | AHP-I | 792.6 $\pm$ 3.35 | 32.47 ms |
> | AHP-V | 764.3 $\pm$ 10.8 | 31.21 ms |
>
> Both AHP variants reduce latency while drastically outperforming the MPPI baseline (BOOM, 326.0), but yield slightly lower TAR than our original parallel MLAP. This suggests that adaptive triggering is a promising way to improve efficiency, although MLAP remains stronger when prioritizing performance.
>
> (Note: All results are averaged over 3 seeds).
>
> # Final thanks
> Thank you again for your time, effort, and professionalism. We hope our responses address your concerns. We are happy to provide additional details if needed, and we look forward to further discussion.

---

> > ### Author Rebuttal · Reviewer_8cYq · 2026-04-04
> >
> > Most of my concerns has been addressed and I am happy to raise the score to 5. I am especially glad to see my Q3 is
> > beneficial and answered fully by the new set of experiments. I encourage authors add this discussion in the newer version of manuscript.

---

> > > ### Author Response · Authors · 2026-04-05
> > >
> > > Thank you very much for your thoughtful follow-up and positive feedback. We will include the discussion and experimental results regarding the Adaptive Hybrid Planning (AHP) in the final camera-ready version of the manuscript. Thank you again for your highly constructive critiques, which have significantly strengthened the comprehensiveness of our paper.

---

### Official Review · Reviewer_c4nK · 2026-03-20

**Soundness:** 3
**Presentation:** 3
**Significance:** 3
**Originality:** 3
**Overall Recommendation:** 5
**Confidence:** 2

**Summary:**

The paper addresses the problem of Model based Reinforcement Learning via a planning-driven method. These kinds of methods use the learned dynamics as a world model for online planning, giving as a result an action sequence at every step. Traditionally, previous methods have relied on Model Predictive Path Integral (MPPI), an online planner that refines actions entirely through sampling. In the proposed approach, authors replace MPPI with Langevin Rollout Optimization (LaRO), a gradient-informed method that leverages information about the model to improve action optimization. The paper makes use of BOOM, a state of the art algorithm, to compare the empirical performance of this approach with MPPI vs LaRO.

**Compliance With Llm Reviewing Policy:**

Affirmed.

**Ethical Review Concerns:**

None.

**Key Questions For Authors:**

The implications of Assumption A.5 are not fully clear to me. How realistic is it to assume bounded estimators? Given that $f$
is in practice a neural network, how can this be ensured? The same happens in Theorem 4.1 with the assumption of bounded score estimation error. This score estimation is done by a neural network, and it seems hard to control. How would you justify this assumption? I would like to see a more extensive discussion of this in the text.

**Limitations:**

I believe the paper would benefit from a more detailed discussion of its limitations and future research. What are the drawbacks of using these gradients? Under what circumstances might they be counterproductive with respect to MPPI? What can we do to build better algorithms using these types of methods?

**Strengths And Weaknesses:**

Strengths:
Figure 1 gives a nice intuition of the effectiveness of the gradient-informed updates.
The motivation and background are clear.
The theoretical contribution of the paper is significant given the complexity of the setting.
The ideas presented in the paper are novel and interesting. The use of Langevin dynamics has been proven to lead to efficient solutions for other reinforcement learning problems in recent years, and I believe further research in this direction is relevant.

Weaknesses:
The experimental results shown in Figure 2 do not seem to confirm a significant and consistent improvement with respect to the ablation (BOOM). In Table 1, the results do show higher average returns for the proposed algorithm. However, the difference with respect to BOOM is not always significant due to the high standard deviation of the latter. This fact also makes me wonder about the reason for the higher standard deviation of BOOM compared to BOOM-L. Is this due to a fundamental difference in the algorithm, or is it due to the experimental setting?

The paper is very heavy on notation, which, on the other hand, seems inevitable given the nature of the setting. However, I find some of the notation confusing. For instance, $q_t$ is introduced in the preliminaries as a terminal value function, while $q_k$ denotes the distribution induced by $a_t$ in Theorem 4.1.

BOOM-L is slightly slower than its MPPI-only counterpart. This deterioration, which I understand stems from the gradient calculation, is not very significant, but it should be taken into account when deciding which algorithm to implement in a real-world scenario.

---

> ### Author Rebuttal · Authors · 2026-03-31
>
> We sincerely appreciate your constructive feedback. Below are our responses to your comments.
>
> # W1: Improvement Significance and Variance vs. BOOM
> We clarify that these differences stem from **fundamental algorithmic differences** rather than the experimental setting.
> - Magnitude of improvement: BOOM is a highly competitive SOTA baseline near the performance ceiling on many tasks. Our goal is to push this boundary further by exploiting the world model's differentiability. While the margin may appear smaller on simple tasks, BOOM-L yields substantial gains on difficult tasks where zero-order sampling struggles. For instance, BOOM-L outperforms BOOM by a remarkable +147.4% on the H1hand-run task.
> - Difference in variance: BOOM's high standard deviation stems from its reliance on purely random MPPI sampling. If initial samples miss the optimal basin in non-convex landscapes, MPPI fails to find a good solution, leading to high variance across seeds. Conversely, BOOM-L utilizes first-order gradients to actively pull the search toward local optima, reducing dependence on initial guesses. This lower variance is empirical evidence that incorporating gradient information yields higher returns and more stable optimization.
>
> # W2: Notation Ambiguity ($q_t$ vs. $q_k$)
> Thank you for pointing out this notation overlap. We update the notation for the terminal value function in the preliminaries from $q_t$ to $Q_t$, aligning with standard RL conventions. We will thoroughly review all mathematical notations in the revision.
>
> # W3: Trade-offs in Real-World Deployment
> The slight inference latency increase is a practical trade-off. In real-world deployments, the choice between BOOM and BOOM-L depends on task requirements:
>
> - When to choose BOOM-L: For highly complex, non-convex tasks requiring precise local coordination (e.g., dexterous manipulation, traversing rugged terrain). In these scenarios, spending an extra ~5 milliseconds on gradient-guided refinement is worthwhile to prevent failures and increase success rates.
>
> - When to choose BOOM (MPPI-only): For simpler, highly dynamic tasks where high-frequency reactive control is the priority (e.g., basic obstacle avoidance), and precise optimality is less critical.
>
> We will explicitly add this discussion to the revised appendix.
>
> # Q1: Realism of Theoretical Assumptions
>
> Our network architecture incorporates standard stabilization techniques widely used in modern MBRL (e.g., TD-MPC2), including Layernorm and SimNorm activations. These techniques help keep the learned representation constrained and the network scale stable.
>
> For **Bounded Jacobians** (Assumption A.5): SimNorm applies a softmax over feature chunks, keeping the latent representation bounded within $(0, 1)$ and guaranteeing a uniformly **bounded Jacobian** (spectral norm $\le 1$). Combined with LayerNorm, which normalizes feature variance to prevent exponential scaling, the overall network Jacobian is controlled in practice. While this structural design does not equate to a strict global Lipschitz bound of 1.0, it effectively prevents gradient explosion, making the bounded-Jacobian assumption a reasonable working condition for our theoretical analysis.
>
> For **Bounded score errors** (Assumption in Theorem 4.1): Theorem 4.1 assumes the expected score matching error is bounded, which aligns directly with our training objective. The score network $S(z, a)$ operates on a strictly **compact input domain**, as it takes naturally bounded actions $a$ and SimNorm-bounded latents $z$ as inputs. According to the Universal Approximation Theorem, a neural network can approximate regular functions arbitrarily well on a compact domain. Furthermore, [Figure 1](https://anonymous.4open.science/r/ICML26_BOOML_rebuttal/Figure1-score_loss.pdf) shows that the actual score-matching training loss remains stable at a consistently low value. Together, these theoretical properties and empirical observations make the bounded-error assumption practical for our analysis.
>
> # Limitations
> We will add a "Limitations and Future Work" section in the revision:
>
> A primary limitation is the trade-off between gradient-guided precision and computational efficiency. Additionally, in scenarios with localized model inaccuracies, pure LaRO might greedily exploit errors and get trapped in local optima, making MPPI's broad stochastic sampling a necessary safeguard in our MLAP framework. Our future work will explore integrating epistemic uncertainty into the score network to explicitly penalize unreliable gradients. Furthermore, developing adaptive triggering mechanisms (e.g. the AHP variant prototyped for Reviewer 8cYq) represents a highly promising path to dynamically balance efficiency and precision.
>
> # Final thanks
> Thank you again for your time, effort, and professionalism. We hope our responses address your concerns. We are happy to provide additional details if needed, and we look forward to further discussion.

---

> > ### Author Rebuttal · Reviewer_c4nK · 2026-04-06
> >
> > My concerns have been fully resolved by the authors. The problems discussed will be updated in the camera ready version of the paper. I would like to thank the authors for their comments and revisions to this paper.

---

> > > ### Author Response · Authors · 2026-04-07
> > >
> > > Thank you very much for your thoughtful follow-up and positive feedback. We will ensure that the clarified theoretical assumptions, corrected notations, and the expanded discussions on practical trade-offs and limitations are fully incorporated into the final camera-ready version of the manuscript. Thank you again for your highly constructive critiques, which have significantly strengthened the clarity and depth of our paper.

---

### Decision · Program_Chairs · 2026-04-30

**Decision:**

Accept (regular)

**Comment:**

This paper proposes a novel algorithm for model-based RL in continuous control. The main contribution is to introduce gradient-guided Langevin rollout optimization together with a score-augmented world model and integrate them into the BOOM planning framework. All reviewers agree this is a good paper. The authors have addressed all the relevant concerns during the rebuttal phase, which included additional training cost analysis and hyperparameter robustness ablations, expanded experimental evaluation with more tasks and seeds, and clarification of the theoretical assumptions and practical limitations. I therefore suggest acceptance.

Please include the following points in the revised version:

- a clear discussion of the training and inference cost trade-offs of BOOM-L relative to BOOM and the variant without the score network.
- the new experimental results added during rebuttal: expanded benchmark and the adaptive hybrid planning analysis.
- a clear presentation of the limitations: especially sensitivity to model errors and the conditions under which gradient-guided planning may be less beneficial.

Regarding references, I found at least an error and several cases where preprints are cited despite the existence of peer-reviewed versions. In particular, the entry "Campbell et al., Model based reinforcement learning for Atari, ICLR 2019" appears to be incorrect, and likely corresponds to "Kaiser et al. (ICLR 2020)", which is already redundantly cited as arXiv preprint. Moreover, multiple works are cited as arXiv preprints even though published versions are available, including Hafner et al. (Dream to Control, ICLR 2020), Hafner et al. (Mastering Atari with Discrete World Models, ICLR 2021), Hansen et al. (Temporal Difference Learning for MPC, ICML 2022), Hansen et al. (TD-MPC2, ICLR 2024), Micheli et al. (Transformers are Sample-Efficient World Models, ICLR 2023), and Sferrazza et al. (HumanoidBench, RSS 2024), among others. These issues should be corrected.